# Coupled oscillations enable rapid temporal recalibration to audiovisual asynchrony

Therese Lennert [1✉], Soheila Samiee[1] & Sylvain Baillet [1✉]

The brain naturally resolves the challenge of integrating auditory and visual signals produced by the same event despite different physical propagation speeds and neural processing latencies. Temporal recalibration manifests in human perception to realign incoming signals across the senses. Recent behavioral studies show it is a fast-acting phenomenon, relying on the most recent exposure to audiovisual asynchrony. Here we show that the physiological mechanism of rapid, context-dependent recalibration builds on interdependent pre-stimulus cortical rhythms in sensory brain regions. Using magnetoencephalography, we demonstrate that individual recalibration behavior is related to subject-specific properties of fast oscillations (>35 Hz) nested within a slower alpha rhythm (8–12 Hz) in auditory cortex. We also show that the asynchrony of a previously presented audiovisual stimulus pair alters the preferred coupling phase of these fast oscillations along the alpha cycle, with a resulting phase-shift amounting to the temporal recalibration observed behaviorally. These findings suggest that cross-frequency coupled oscillations contribute to forming unified percepts across senses.

[1] McConnell Brain Imaging Centre, Montreal Neurological Institute, McGill University, Montreal, QC, Canada. ✉email: therese.lennert@mail.mcgill.ca; sylvain.baillet@mcgill.ca

The temporal coincidence of natural stimuli in different sensory modalities provides an important ecological cue for determining which stimuli should be bound together to form a unified percept. Yet, optimal integration is challenged by the fact that most cross-sensory stimulus pairs have different physical transmission and physiological transduction times. This results in discrepancies in relative processing latencies in the order of tens of milliseconds[1]. To ensure coherent perception of the world, the brain perceptually realigns corresponding inputs between the senses—a mechanism known as *temporal recalibration*[2,3].

For audiovisual stimulus pairs, temporal recalibration has been shown to occur after prolonged adaptation to a fixed audiovisual asynchrony[2,3] and more recently, also on a trial-by-trial basis contingent only upon the asynchrony on the preceding trial[4,5]. For example, the presentation of an audiovisual stimulus asynchrony of 100 ms makes subsequent asynchronies of similar duration appear less pronounced than before. Such rapid temporal recalibration allows for the instantaneous implementation of small, modality-specific shifts in temporal processing thereby facilitating multisensory integration for unified perception. Such a fast time scale would be particularly beneficial when processing dynamically changing stimuli e.g., during speech comprehension[6]. Emerging evidence suggests that rapid temporal recalibration is altered in individuals on the Autism Spectrum, which is associated with abnormalities in multisensory temporal processing[7,8].

While recent studies have clarified the neural processes of temporal recalibration after prolonged adaptation to asynchronous stimulus pairs[9,10], very little is known about the neural mechanisms underlying rapid temporal recalibration. Simon and co-workers demonstrated using EEG that the magnitude of neural responses to simple audiovisual stimuli was modulated by the temporal order of the stimuli on the previous trial[11]. When there was a mismatch—e.g., visual lead on the previous trial followed by auditory lead on the current trial—late components of the evoked potentials over centro-parietal brain regions were larger compared to successive trials with identical temporal orders. In a set of experiments using speech events, they further provided a link between the magnitude of the observed neural modulations and ongoing decisional dynamics[12]. The authors concluded that late, higher-order processes are the main contributors to rapid temporal recalibration behavior.

It is to date unclear how rapid temporal recalibration affects early sensory processing and the neural dynamics of brain systems. Because the order of the sensory modalities on the previous trial influences synchrony perception on the subsequent trial, we postulated that neural signatures of rapid temporal recalibration shall manifest during the inter-trial interval to optimize temporal processing of the upcoming stimulus pair[13,14]. To describe the neural mechanisms underlying such rapid temporal recalibration, we focused on rhythmic brain activity in sensory regions. We also considered that neural oscillations at different frequency bands interact with each other[15,16]. One subtype of such cross-frequency coupling is phase-amplitude coupling (PAC), whereby low-frequency oscillations (frequency for phase fP) modulate the amplitude of higher-frequency signals (frequency for amplitude fA). PAC is actively researched as a possible mechanism of information encoding and integration by neural assemblies and networks. Generally, it is assumed that high-frequency oscillations are signal markers of local computations, while low-frequency components would contribute to signal integration across larger distances[17,18]. Similar to the functional role that PAC plays in the hippocampus[19–22], it has been suggested that cortical PAC provides a temporal segmentation mechanism that discretizes continuous stimuli into smaller

chunks for further processing[23,24]. We embrace this idea and put forward the hypothesis that fast oscillations typically in the human beta/gamma bands (25–100 Hz) may be related to discrete slots, or opportunities for the brain to register the neural representations of incoming stimuli in a temporally organized manner along the cycles of slower e.g., alpha rhythms (8–12 Hz)[25].

Specifically, we propose that PAC in auditory and visual cortices is a key mechanistic component for audiovisual sensory events to be registered as occurring simultaneously and processed as such further downstream by higher-order circuits to adjust behavior. One possible mechanistic implementation would be that stimulus processing is shifted to a previous or subsequent slot—e.g., by one high-frequency fA oscillatory cycle or more—thereby determining in a flexible manner the magnitude of temporal recalibration that takes place. Temporal recalibration behavior after audiovisual asynchrony is subject-specific with an average extent of 30–40 ms[4]. Fast cortical oscillations within the beta and gamma ranges have short periods and could indeed provide the functional flexibility necessary at the short time scales observed behaviorally in temporal recalibration. We thus tested whether signatures of regional neural processing—i.e., high-frequency oscillations coupled to the phase of a slower rhythm—prior to stimulus onset in the auditory and visual cortex, would provide a mechanistic framework accounting for the individual temporal adjustments observed behaviorally in simultaneity perception after audiovisual asynchrony. We used magnetoencephalography (MEG) source imaging in human participants performing a simultaneity judgment task, in which an audiovisual stimulus pair was presented in different temporal configurations (Fig. 1).

## Results

**Temporal recalibration occurs rapidly contingent upon the previous trial.** To demonstrate temporal recalibration, we pooled data according to the order of modality presentation on the previous trial (t-1); i.e. current trials (t) preceded by a visual-lead presentation (t-1:V condition) vs. an auditory-lead presentation (t-1:A condition). We computed the percentage of synchronous responses as a function of SOA and fitted a Gaussian model to the

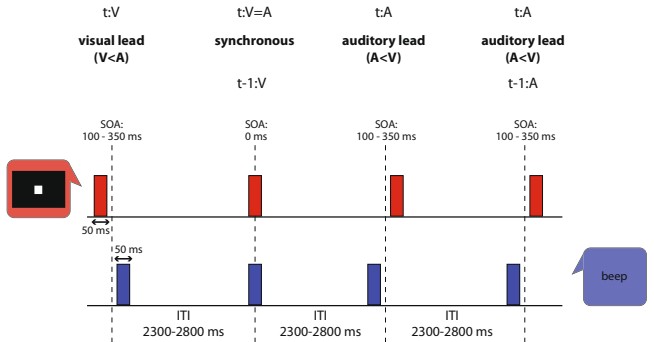

**Fig. 1 Trial types and behavioral task.** Time course of an example stimulus sequence used to test audiovisual synchrony judgments (illustration adapted from[2]). The audiovisual stimulus pair was presented in one of three possible temporal configurations: a visual stimulus (red) leading an auditory presentation (blue; t:V, V < A; t represents current trial), synchronous audio-visual presentation (t:V=A), and auditory leading visual (t:A, A < V). To study rapid temporal recalibration effects, we capture the context of a given trial t by indicating the type of the preceding trial (t-1) as e.g., t-1:V, as illustrated here with the synchronous presentation trial t:V=A. After each stimulus pair, the participant had to indicate by button press whether the two stimuli were synchronous or not. Seven levels of Stimulus Onset Asynchrony (SOA) were used ranging from 0 to ±350 ms. The duration of the inter-trial interval (ITI) ranged between 2300 and 2800 ms.

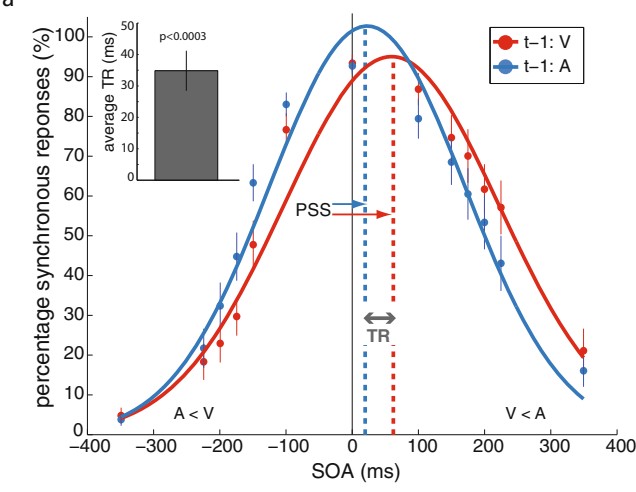

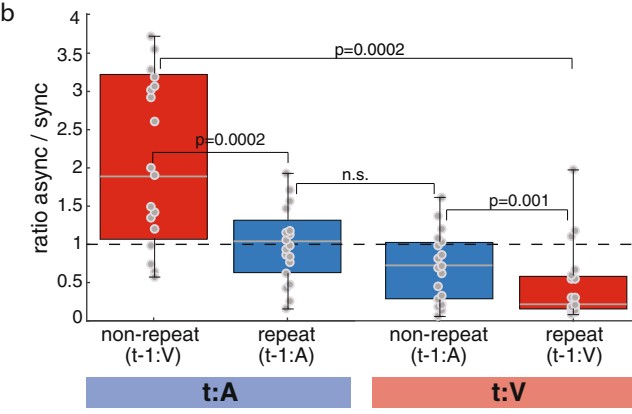

**Fig. 2 Behavioral performances. a** Synchronous responses as a function of stimulus onset asynchrony. Psychometric curves (Gaussian model adjusted to behavioral measures) showing the percentages of synchronous responses as a function of SOA for trials with visual lead on the previous trial (t-1:V, red) and those with auditory lead (t-1:A, blue). Dots represent average behavioral reports across participants (n = 18). The colored thin lines indicate the standard errors of the mean (SEM). Inset: mean temporal recalibration (TR) estimate and SEM across participants evaluated against zero with a two-tailed, one-sample t test. **b** Individual ratios of asynchronous-to-synchronous responses for all trial configurations. Ratios of asynchronous to synchronous responses across participants for all t-1/t trial combinations, i.e. auditory or visual lead on trial t (t:A or t:V) paired with visual or auditory lead on the previous trial (t-1:V and t-1:A). The lower and upper limits of the red and blue boxes represent the interquartile range (quartiles Q1 and Q3); gray center lines denote Q2 (median); whiskers represent minimal and maximal data points; gray dots represent single-subject data (n = 18). The dashed black horizontal line highlights a ratio of 1. Wilcoxon signed-rank tests were performed between all trial combinations; n.s.: p > 0.0125 after accounting for multiple comparisons.

data (Fig. 2a). The SOA histogram mode corresponds to the Point of Subjective Simultaneity (PSS), defined as the point where two different sensory inputs are perceived as maximally simultaneous. The PSS values were shifted toward visual leads (550 ± 13 ms SEM in the t-1:V condition and 21 ± 12 ms SEM in the t-1:A condition), an asymmetry replicating previous reports[26]. The PSS values between the two study conditions were significantly different (paired-sample t test, t = 5.28, p = 0.00009). Rapid temporal recalibration is a measure of how much the PSS is shifted between t-1:V and t-1:A trials. In other words, temporal recalibration describes by how much simultaneity perception is recalibrated after a given asynchrony exposure. Temporal

recalibration quantifies the amount of such shift and is defined as the difference between the PSS of the t-1:V condition and the PSS of the t-1:A condition. We found that on average, participants dynamically recalibrated their perception of simultaneity by 35 ms (±6 ms SEM; one-sample t test against zero, t = 2.11, p = 0.0003, confidence interval = [32.03 37.98]; Fig. 2a inset), an observation in agreement with the original findings by Van der Burg et al. [4].

**Valid expectation of sensory modality order increases synchronous perceptions.** We also obtained the participants' ratios of asynchronous-to-synchronous responses for all trial configurations (Fig. 2b). For both auditory leads and visual leads on the current trial, a repeat of sensory modality order between two consecutive trials significantly reduced the ratio of asynchronous-to-synchronous responses (t:A, median 1.88 (non-repeat, red) and 1.03 (repeat, blue), Wilcoxon signed-rank test, p = 0.0002; t: V, median 0.67 (repeat, blue) and 0.25 (non-repeat, red), p = 0.001). This result corroborates a central notion to temporal recalibration: previous exposure to a given amount of audiovisual asynchrony on trial t-1 causes a shift in PSS, such that similar asynchronies on next trial t appear less pronounced—i.e., the current pair of stimuli is perceived as more synchronous than before[2–5,11,12]. The data further revealed that significant changes in the ratio of asynchronous-to-synchronous reports occur only following visual leads on the previous trial (median 1.88 (t:A, red) vs. 0.25 (t:V, red), Wilcoxon signed-rank test, p = 0.00021). Following auditory leads on trial t-1, ratios were similar for auditory and visual leads on the current trial (median 1.03 (t:A, blue) vs. 0.67 (t:V, blue); p = 0.1989). This finding suggests an asymmetry in the direction of temporal recalibration in our task.

**Phase-amplitude coupling in auditory and visual cortex.** We extracted measures of PAC within the sensory regions maximally activated by the presentation of the audiovisual stimulus pairs (four functionally localized regions of interest (ROIs): left and right auditory cortex, LAC/RAC; left and right visual cortex, LVC/RVC; Fig. 3a). Prominent spectral peaks in the alpha range in all ROIs (Fig. 3b, bottom right panel) pointed at this band as a candidate for low-frequency range for phase (fP) in subsequent PAC analyses. Further, because a well-defined spectral peak is necessary for the meaningful estimation of PAC[27], 2 out of 18 participants were excluded from PAC analyses (Supplementary Fig. 1). Because fast cortical oscillations within the beta and gamma ranges have short periods compatible with the subject-specific temporal recalibration observed behaviorally, frequencies in the range of 16–84 Hz defined frequency bands of interest for a possible relation with PAC's frequency for amplitude (fA).

To provide empirical evidence of PAC, we replicated the approach of Canolty et al.[28] and computed time-frequency maps of neurophysiological MEG source activity over 1-s epochs, for each participant and ROI, time-locked to the troughs of the subject-specific regional fP oscillations[28,29]. Figure 4a displays for one example participant strong manifestation of PAC between fP at 10 Hz and fA in the range of 50–130 Hz for all ROIs. For all participants, PAC was confined to relatively narrow ranges of fA frequencies, with no spectral peaks at harmonic frequencies of fP, arguing against spurious forms of coupling that could have been caused by harmonics of non-sinusoidal fP rhythmic fluctuations. We assessed the statistical significance of PAC during the pre-stimulus time period (−500 ms to 0 ms) using a non-parametric prevalence procedure based on surrogate datasets with no PAC above chance levels[27]. We determined on a single-trial basis the combination of fP and fA that yielded the maximum PAC strength[30] (see "Methods" for details). For each participant and

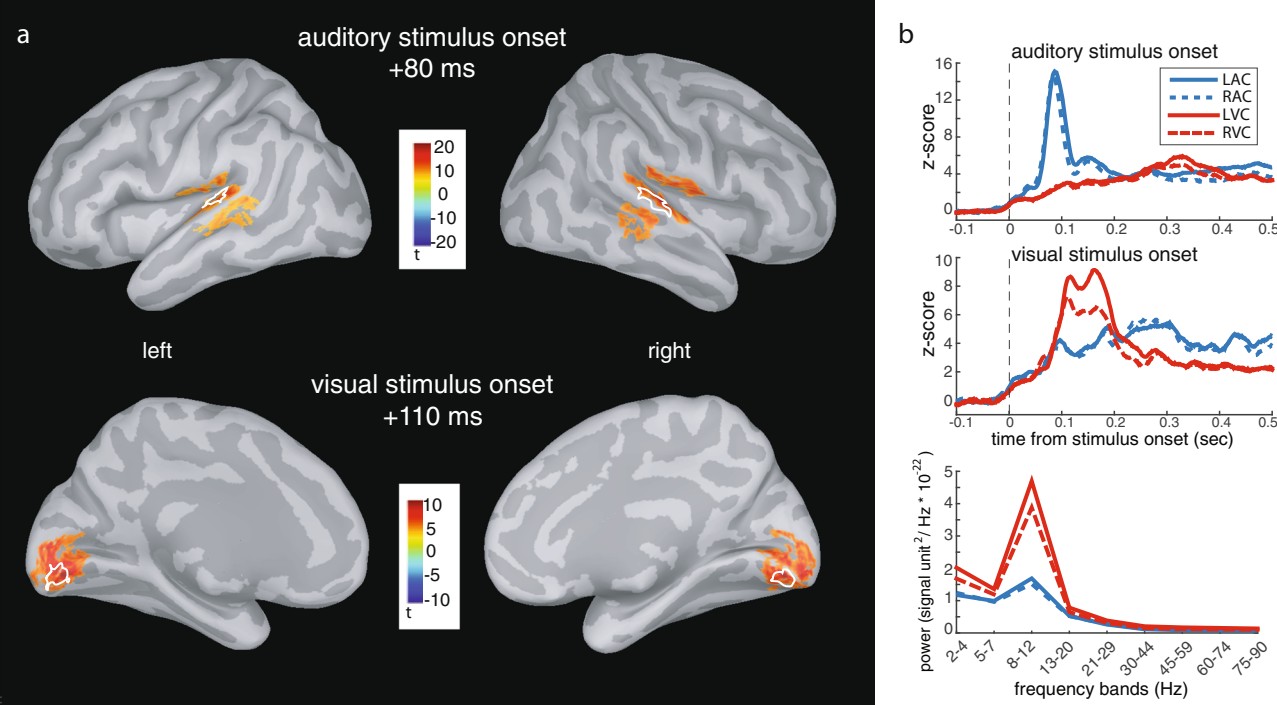

**Fig. 3 Regions of interest: anatomy and basic neurophysiological responses. a** MEG cortical source modeling of event-related brain responses. Group average ($n = 18$) event-related brain responses to auditory stimuli initiated around 50 ms and were maximal at 80 ms after auditory stimulus onset and localized to left and right superior temporal gyri (LAC/RAC). Responses to visual stimuli peaked on average 110 ms post-stimulus and localized to left and right striate and extrastriate visual cortices (LVC/RVC). White outlines represent the regions of interest in one representative participant. **b** Region of interest time series and power spectrum densities. The group average source time courses depict activation across time extracted from the participants' ROIs for auditory and visual stimulus onsets (top and middle panels). Power spectrum density estimated over the pre-stimulus time period of all trials, across the indicated frequency ranges and for all ROIs (bottom panel).

each trial, PAC levels from task data and surrogate data were compared using their z-scores (Supplementary Table 1). Positive Z-scores indicate higher PAC scores for task trials. Across participants, Z-scores were strictly positive in 69–75% of the trials in all ROIs, indicative of a strong trend away from the surrogate data (Fig. 4b). Evaluation of the average Z-scores (LAC: $Z = 1.64$, RAC: $Z = 1.54$, LVC: $Z = 1.82$, RVC: $Z = 1.72$) against the critical value of $Z = 2.24$ (corresponding to the Bonferroni-corrected significance level (alpha) of 0.0125 in an upper-tailed Z-test), confirmed trends in all ROIs. A possible reason for the fact that pre-stimulus PAC strength remained below statistical significance after Bonferroni correction in auditory and visual cortex is the absence of any sensory input and stimulation at this point during the task: Participants were anticipating the presentation of a new stimulus pair but could not anticipate the timing of its occurrence. However, visual inspection of the data in time-frequency maps (as in Fig. 4a) and the dismission of spurious PAC due to broadband/harmonics coupling strongly advocates in favor of actual expressions of PAC in the empirical traces. Overall, we found across all trials and participants that fP (respectively fA) was on average 9.6 Hz ± 0.02 Hz SEM) (respectively, 25.6 Hz ± 2.5 Hz SEM) in auditory ROI and 9.7 Hz ± 0.03 Hz SEM (respectively, 24.8 Hz±1.6 Hz SEM) in visual ROI, with no significant difference between ROIs ($p = 0.30$, respectively $p = 0.86$, paired $t$ tests from MATLAB's Statistics & Machine Learning Toolbox version 11.5, MATLAB version 9.6).

**Temporal recalibration relates to the period of coupled fast-frequency oscillations.** Based on the previously proposed idea that PAC may provide a form of temporal segmentation mechanism that discretizes continuous stimuli into smaller

chunks or slots, for further processing[23–25], we hypothesized that rapid temporal recalibration is enabled by phase shifts of fA nested oscillatory segments (representing the "slots") along the slower fP oscillations. Depending on the asynchrony of the stimulus pair on the preceding trial, nested high-frequency cycles would be advanced or delayed along the underlying slow cycle. Consequently, stimulus processing would be shifted to a previous or subsequent slot—e.g., by one high-frequency fA oscillatory cycle or more—thereby determining in a flexible manner the magnitude of temporal recalibration that takes place. Accordingly, we first assessed whether subject-specific temporal recalibration behavior was related to the period ($T_A$) of the subject-specific fA fast oscillations (i.e. $T_A = 1/fA$) measured prior to stimulus onset ($-500$ ms to 0 ms). We applied robust linear regression analysis between the participants' individual fA period and their empirical temporal recalibration measures.

We observed in both auditory cortices, strong positive interactions between temporal recalibration performance and the fA period for visual leads on the previous trial. The period of the fast subject-specific fA oscillation increased as temporal recalibration increased (t-1:V, $p = 0.0004$ (LAC) and $p = 0.0005$ (RAC); evaluated at $p < 0.0042$ after multiple-comparison correction (12 comparisons); Fig. 5). The slopes of the regression lines (parameter $a$) were on average 1.49 ($a = 1.58$ and $a = 1.39$), indicating that the amount of recalibration approached 1.5 times the duration of a fast fA cycle. The effect sizes were large with $r^2$ of 0.78 and 0.77. For auditory leads on the previous trial, we observed positive interactions between recalibration and the fA period, which remained a trend at the corrected alpha level in both auditory cortices (t-1:A, RAC: $p = 0.0046$, $a = 0.88$, $r^2 = 0.55$; LAC: $p = 0.0259$, $a = 0.96$, $r^2 = 0.67$). This finding indicates a stronger linear relationship for visual leads on the previous trial

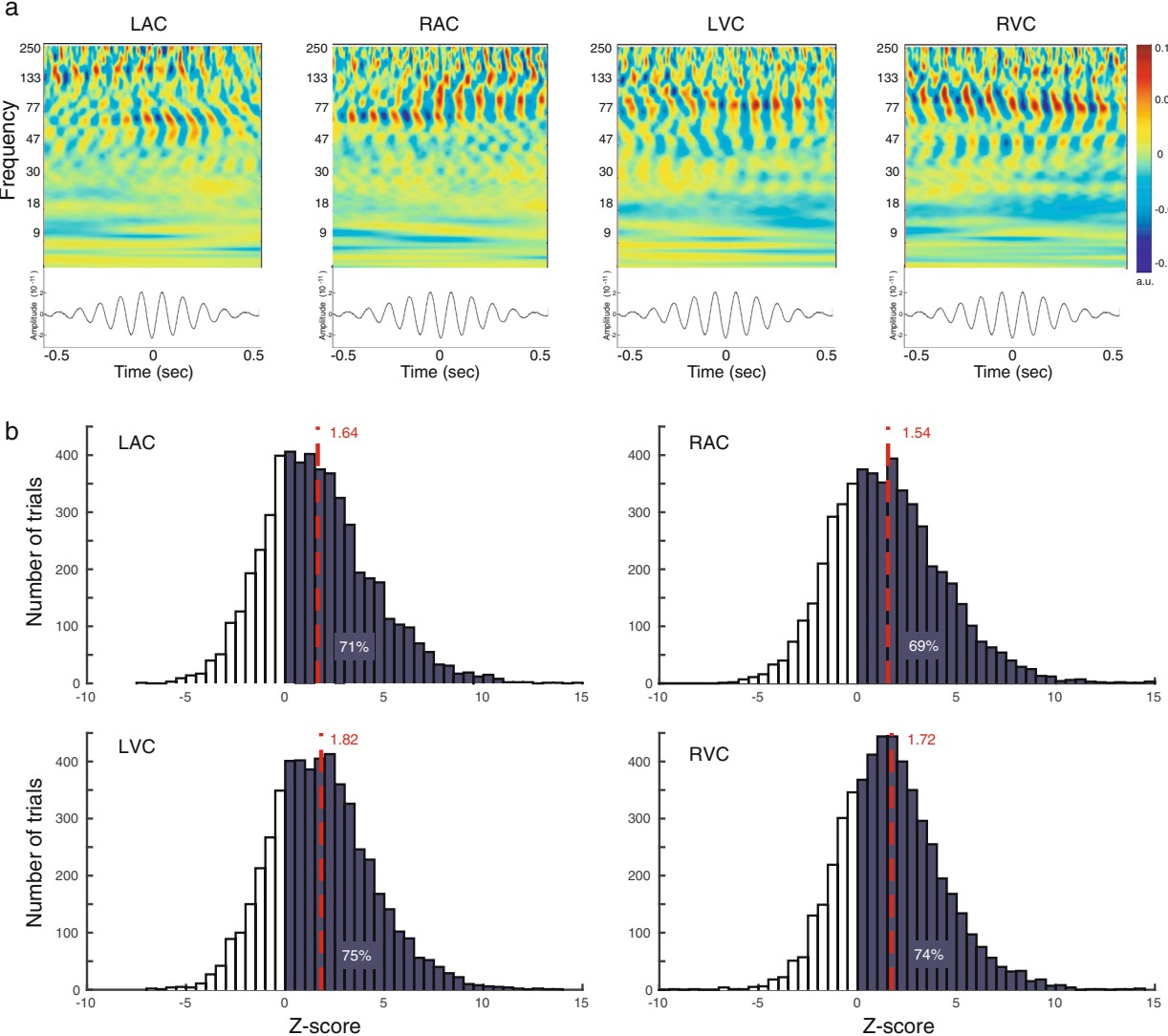

**Fig. 4 Inter-trial, pre-stimulus PAC in auditory and visual cortex. a** Empirical evidence of PAC manifestations in neurophysiological cortical traces. Time-frequency decompositions (top) of neurophysiological traces and their event-related average (bottom) time-locked to the troughs of slow-frequency fP (10 Hz) cycles, for each ROI in a representative participant. LAC, RAC: Left/right auditory cortex; LVC/RVC: Right auditory cortex. **b** Statistical significance of measured PAC effects. Group average (n = 16) histograms of z-transformed PAC measures of task data across trials, relative to surrogates. Dark-shaded areas emphasize positive task PAC z-scores (where PAC was stronger than in surrogates) with their prevalence across trials in each region indicated as percentages. The dashed red lines indicate the mean task PAC z-score in each region, averaged across trials and participants.

which is in accordance with the behavioral results above (see Fig. 2b). Together these data suggest that t-1:V trials may be more strongly temporally recalibrated than their t-1:A counterparts. In the visual cortex, no significant correlations were observed during the pre-stimulus period (all $p > 0.05$, mean $a = 0.78$, mean $r^2 = 0.4$; see Supplementary Fig. 2a). Likewise, we did not find significant interactions over the post-stimulus time period (250–750 ms after stimulus onset; all $p > 0.01$, mean $a = 0.86$, mean $r^2 = 0.4875$; Supplementary Fig. 2b). In sum, our results demonstrate that the period of fA oscillations prior to stimulus onset is quantitatively, and consistently between participants, related to individual temporal recalibration behavior.

Because superior temporal-lobe theta oscillations are prominently reported in auditory processing[23,31], we evaluated whether their period duration would also be linearly associated with individual temporal recalibration behavior. We therefore conducted the same PAC analysis using theta (4–7 Hz) as candidate fP frequency range during the pre-stimulus period. The data did not show a significant association between the period of fA

oscillations coupled with theta as fP and temporal recalibration behavior (all $p > 0.05$, mean $a = -0.64$, mean $r^2 = 0.23$; Supplementary Fig. 2c).

Finally, we verified that the PAC analysis was not confounded by possible differences in the magnitude of alpha-band pre-stimulus brain activity (8–12 Hz) in the auditory cortex between the t-1:A and t-1:V conditions (Supplementary Fig. 3) or differences in PAC strength (Supplementary Fig. 4).

**PAC phase shifts in auditory and visual cortex account for temporal recalibration**. We hypothesized that temporal recalibration behavior is related to phase shifts of the fA nested cycles along the slower fP oscillations. Specifically, we predicted that depending on the asynchrony of the stimulus pair on the preceding trial, nested high-frequency cycles would be advanced or delayed along the underlying slow cycle. In order to test this idea, we extracted the preferred phase φ of fA occurrences in the fP cycle from individual PAC estimates in each ROI and for both

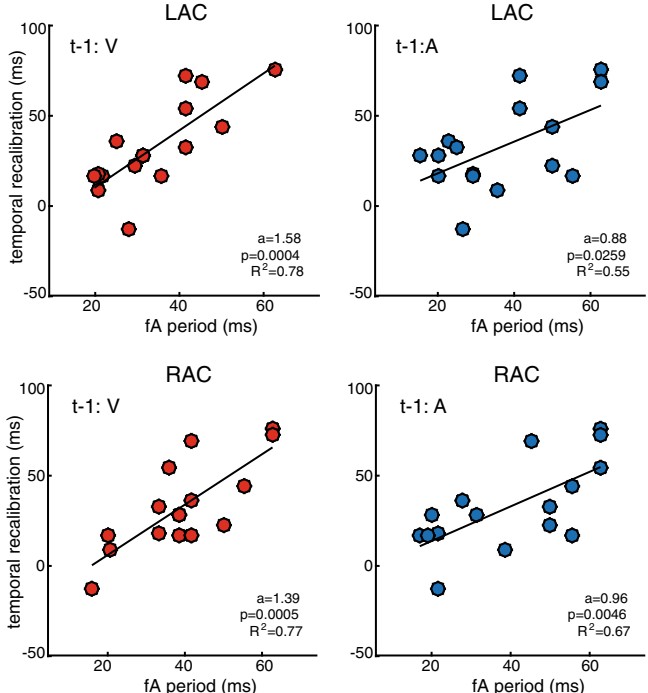

**Fig. 5 Temporal recalibration is quantitatively related to fA period.**
Individual temporal recalibration estimates ($n = 16$) are plotted against the observed fA period durations measured over the pre-stimulus period in the auditory cortex for trials with visual (red) and auditory (blue) lead on the previous trial. The dots represent single-subject values. In each panel, the regression line of best fit is depicted in black. The regression coefficients a, p-values, and effect size $r^2$ are indicated in the respective panels.

trial conditions (t-1:V and t-1:A). We examined whether auditory and visual stimulus leads on the previous trial influenced this preferred coupling phase prior to the next audiovisual stimulus pair. We estimated on each trial and for every cortical source within each ROI the phase in the alpha cycle when amplitude modulation of fA oscillations was the strongest. The resulting phase distributions between t-1:A and t-1:V conditions were significantly different in the right auditory and right visual cortex (Kuiper tests over t-1:A and t-1:V conditions group effects, $p = 0.001$ and $p = 0.005$, respectively; Fig. 6a. Supplementary Fig. 4a provides a single-subject example for the same two conditions, $p = 0.02$ and $p = 0.005$ respectively) indicating that the modality order on the previous trial influenced PAC's preferential coupling phase immediately prior to the presentation of the next stimulus pair. We did not find significant differences in phase distributions in the tested left hemisphere regions (Supplementary Fig. 4b); we thus focused the subsequent analyses on the right auditory and visual ROIs. We selected from the right auditory and visual ROIs the cortical locus with maximum phase difference between the t-1:A and t-1:V conditions in each participant. We then determined the consistency of $\varphi$ across participants for each condition. We observed significant phase angle consistency across participants in the right auditory cortex in the t-1:A condition ($p = 0.0023$, Rayleigh test of uniform distribution of circular measures) and a trend in the t-1:V condition ($p = 0.0307$, evaluated at corrected $p < 0.0125$). This effect was also significant in the right visual cortex in both conditions (t-1:A, $p = 0.0035$ and t-1:V, $p = 0.00001$; Fig. 6b). Remarkably, the mean shifts in preferred phase angles induced by the leading sensory input modality on the previous trial (t-1:A vs. t-1:V) were similar in the auditory and visual cortex: a 145-degree phase shift between conditions in RAC (from

351 degrees for t-1:A to 136 degrees for t-1:V); a 139-degree phase shift between conditions in RVC (from 276 degrees for t-1:A, to 55 degrees for t-1:V). For both conditions and all regions of interest, we converted the phase shifts of the $\varphi$ parameter from radian to milliseconds with

$$\varphi_{[ms]} = 1000 \times \varphi_{[radians]}/(2\pi fP). \qquad (1)$$

Across participants, phase shifts (in milliseconds) were statistically identical to temporal recalibration behaviors (34.4 ms ± 5.6 ms SEM), both in the auditory cortex (32.5 ms ± 3.7 ms SEM; $p = 0.74$) and visual cortex (29.6 ms ± 3.9 ms SEM; $p = 0.57$; Fig. 6c). Robust linear regression analysis revealed an association between the individuals' temporal recalibration and phase shift (in visual cortex, $p = 0.023$, and a trend in auditory cortex, $p = 0.097$; Supplementary Fig. 5). These observations of the preferred phase of fA-to-fP coupling emphasize the account of individual temporal recalibration behavior.

**Proposition of the dynamic insertion model.** Based on the above findings, we propose the *Dynamic-insertion* model of multimodal synchrony perception. Following Jensen et al.[25], the model premise is that the amplitude of alpha oscillations marks the amount of net inhibition in cortical cell assemblies, and that the rapid alternations of nested fast oscillations decrease inhibition momentarily with respect to the ongoing amplitude of the underlying alpha cycle. As inhibition ramps up within a slow cycle but is transiently reduced over one occurrence of the fast cycle, neuronal representations are instantiated sequentially according to their respective excitability. We further propose that such instantiation is behaviorally crucial for perception to assess the relative timing of concurrent incoming inputs to multiple sensory modalities. Each cycle in a nested segment of high-frequency oscillations indexes the temporal ordering of incoming sensory events (illustrated as insertion *slots* s1, s2, s3 in Fig. 7a). Let us consider an audiovisual event consisting of a pair of stimuli from each modality presented sequentially. The Dynamic-insertion model predicts that the two stimuli in the presented pair will be perceived as synchronous if their respective neural representations reach their respective cortical input regions at the occurrence of the same rapid alternation of the local nested fast oscillation—i.e. they are inserted in the same sensory registration slot (e.g., s1 in both the auditory and visual cortex). This neurophysiological scenario corresponds to the PSS perceptually. As the SOA between the auditory and visual stimulus in a pair departs from PSS, the corresponding cortical inputs are inserted into different sensory registration slots (e.g., s1 in visual cortex and s2 in the auditory cortex) and the pair is subjectively perceived as asynchronous. Our present findings indicate that fast temporal recalibration behavior contingent upon the context of previously observed stimulus presentations could be enacted through *dynamic* adjustments of the preferred phase $\varphi$ when nested high-frequency fA oscillatory cycles occur along the slower fP cycle. The Dynamic-insertion model explains that stimulus registration can be shifted to a previous or subsequent sensory registration slot (highlighted with pink arrows in Fig. 7b)—e.g., by one high-frequency fA oscillatory cycle or more—thereby adapting the magnitude of temporal recalibration in a flexible manner, depending on context and namely, prior audiovisual sensory inputs. The subsequent effect of this phase shift is that on the next trial both the auditory and visual stimuli are more likely to be registered to a slot of the same rank. Consequently, the resulting subjective perception of the stimulus pair is more synchronous than on the previous trial.

We derived the computational terms of the Dynamic-insertion model to explain PSS behavior from neurophysiological

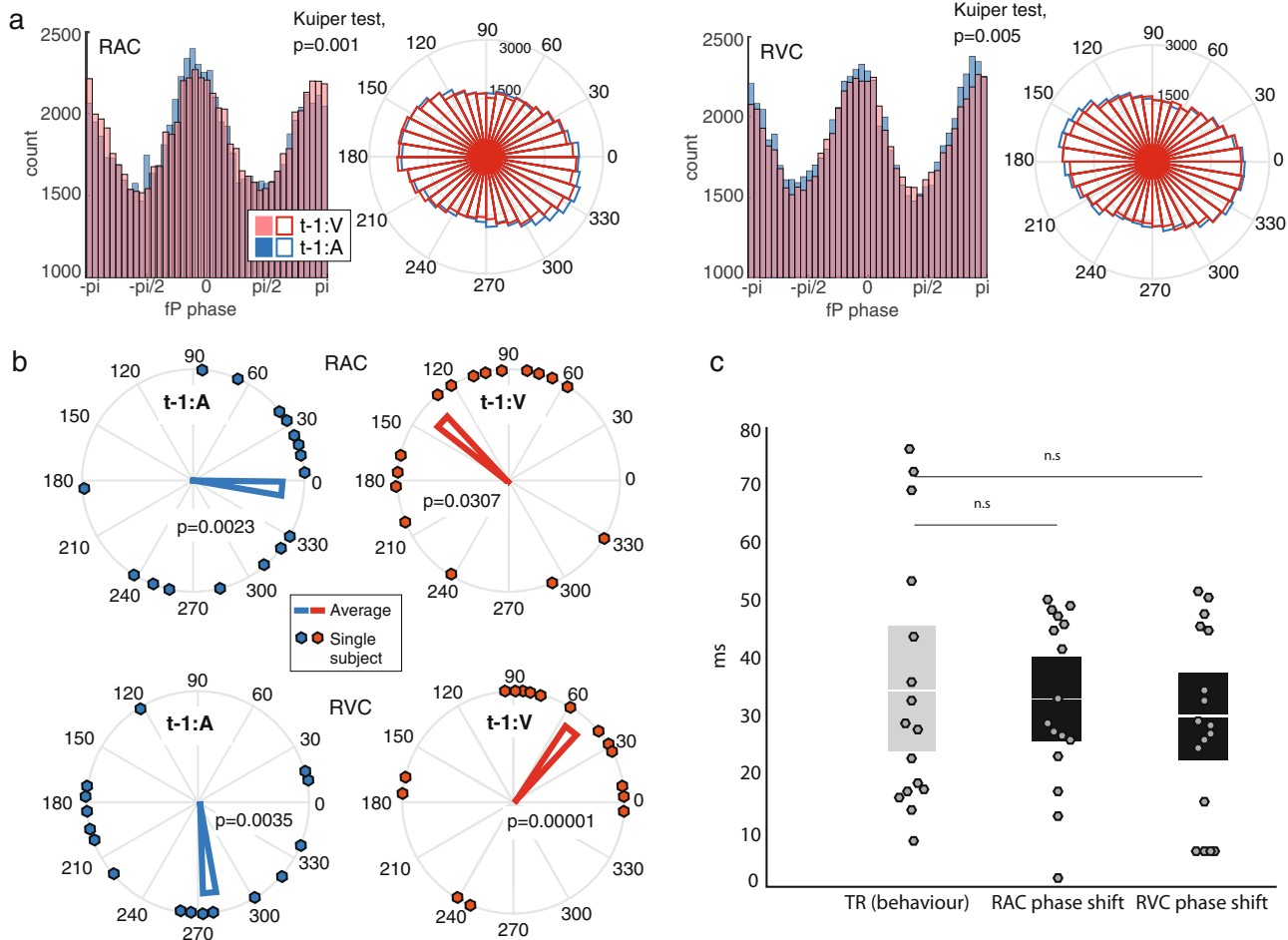

**Fig. 6 Modality order on the previous trial modulates preferred coupling phase. a** Histogram and rose plots show the empirical distribution of single-trial preferred phase angles of PAC along the fP cycle, for trials with visual lead on the previous trial (t-1:V; red) and those with auditory lead (t-1:A; blue) extracted from all MEG sources within RAC and RVC and pooled across participants ($n = 16$). The Kuiper test assessed whether the two circular distributions were significantly different. **b** Rose plots show the distribution of individual preferred phase angles (dots) and the group circular average of PAC preferred phase $\varphi$ of fA oscillations along the fP cycle. Data from trials with visual lead on the previous trial (t-1:V condition) are shown in red and those with auditory lead (t-1:A condition) are shown in blue, in RAC (top) and RVC (bottom). Rayleigh tests assessed uniformity of phase angle distributions across subjects around the unit circle (p-values are displayed). **c** The PAC phase shifts measured between conditions in the auditory (left black bar; RAC) and visual (right black bar; RVC) cortex, once converted to milliseconds from phase angles along the fP cycles (Equation 1), were statistically identical on average to the empirical values of temporal recalibration (gray bar; TR), (both with $p > 0.5$; two-sampled paired $t$ tests). Dots represent single-subject data ($n = 16$).

observations (fA, fP, and $\varphi$). We hypothesized that temporal recalibration behavior is conditioned by the previous audiovisual presentation (trial t-1:V or t-1:A), in anticipation of the subsequent presentation of an audiovisual stimulus pair. We implemented the model considering that $\varphi$ is defined by when along the fP cycle the middle duration of fA oscillatory events tends to occur[30]. We defined as $\alpha$, a scalar multiplier of the $T_A$ period duration of a fA cycle ($T_A = 1/fA$), so that $\varphi$-$\alpha T_A$ is the phase latency of the first sensory registration slot s1 along the fP cycle. For simplicity, we assumed that $\alpha$ is identical in the auditory and visual cortex. At PSS, both incoming sensory signals are registered to the respective s1 slots of auditory and visual cortex, yielding

$$\varphi_A - \alpha T_{AA} = \text{PSS} + 20, \qquad (2)$$

and,

$$\varphi_V - \alpha T_{AV} = 50, \qquad (3)$$

where $\varphi_A$ (respectively $\varphi_V$) is the PAC phase of the occurrence of fA oscillations along the fP cycle, expressed in milliseconds, in the

auditory (respectively visual) cortex; $T_{AA}$ (respectively $T_{AV}$) is the period duration of fA oscillations in the auditory (respectively visual) cortex; PSS is expressed in milliseconds; 20 (respectively 50) is the typical latency of the auditory (respectively visual) sensory signal inputs to cortex, also in milliseconds. Combining the two equations yields an account of PSS behavioral measures from PAC neurophysiological parameters:

$$\text{PSS} = \varphi_A - (\varphi_V - 50)(T_{AA}/T_{AV}) - 20. \qquad (4)$$

The resulting dynamic-insertion model estimates of PSS across participants were statistically identical to the behavioral measures in both the auditory lead t-1:A condition (Dynamic-insertion model predictions: 22.9 ms ± 13.5 ms SEM; Behavior: 21.3 ms ± 13.2 ms SEM; $p = 0.95$, Bayes factor = 3.91) and the visual lead t-1:V condition (Dynamic-insertion model predictions: 51.9 ms ± 10.5 ms SEM; Behavior: 54 ms ± 14.5 ms SEM; $p = 0.92$, Bayes factor = 3.90; Fig. 7c).

For the behavioral data, we observed a significant difference between conditions (PSS t-1:A vs. PSS t-1:V, $p < 1e{-}4$), while the

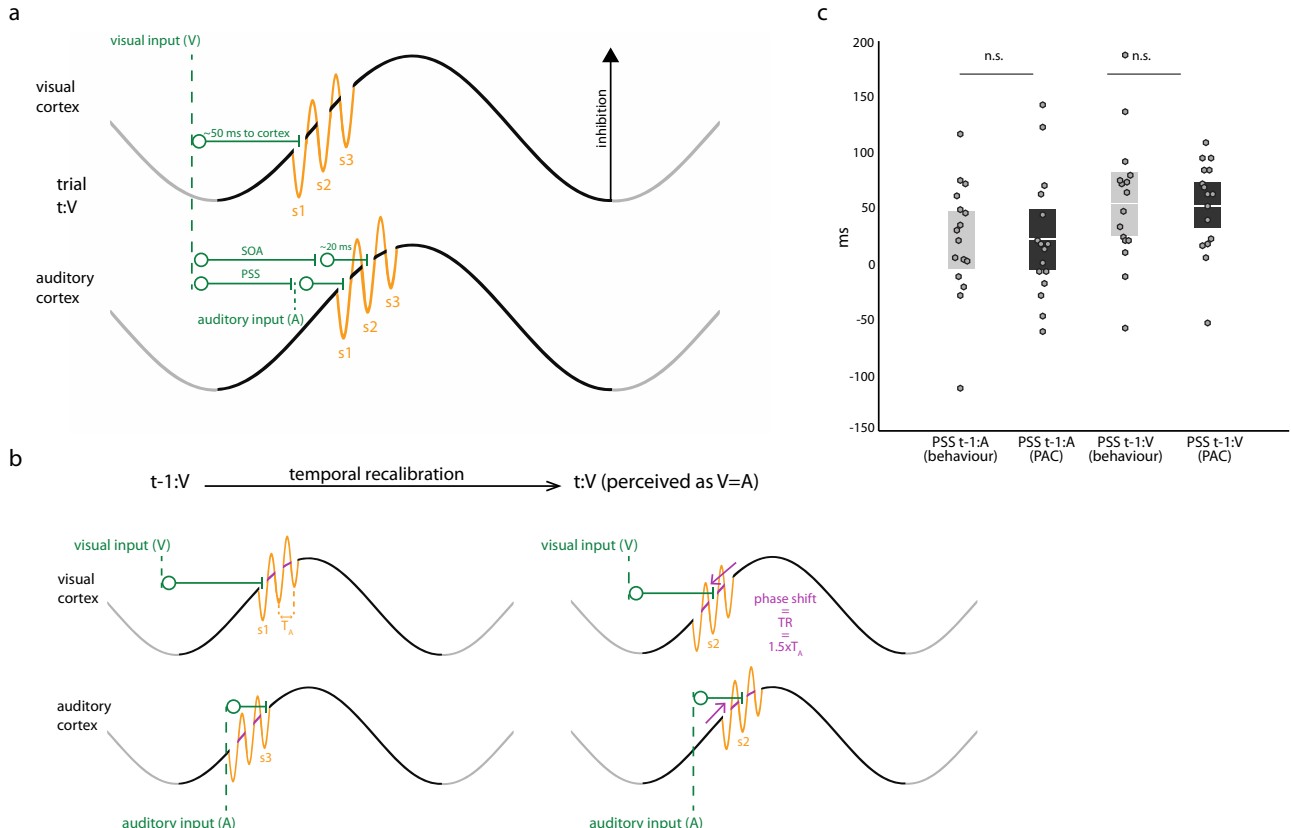

**Fig. 7 Dynamic-insertion mechanism of rapid temporal recalibration. a** Cortical registration of sensory inputs. Schematic illustration of slow oscillatory variations in auditory and visual cortices marking the level of net regional inhibition. PAC manifests as faster oscillatory cycles (orange) nested within slower-frequency cycles. The Dynamic-insertion model posits that each alternation of the fast oscillation defines a sensory registration "slot" (marked as s1, s2, and s3) that transiently reduces inhibition to facilitate the cortical registration of the sensory input. Here we consider the example of a multimodal stimulus pair consisting of a visual sensory input (V, occurrence marked with first green vertical dashed line to the left) preceding an auditory sensory input (A; rightmost vertical dashed line). The trial (t) is labeled t:V to indicate the sequence (V preceding A) of sensory inputs in the pair. The visual neural signal takes about 50 ms to reach the cortex from the retina. The time latency of the auditory stimulus with respect to the visual presentation is the SOA. Neural signals reach the auditory cortex within a considerably shorter amount of time than along visual pathways (~20 ms vs. 50 ms). When the SOA is long enough, the stimulus pair is perceived as asynchronous because the visual cortical input is registered in slot s1 in visual cortex, while the auditory input is registered in slot s2 in auditory cortex. When the SOA is shorter and reaches the PSS, both the auditory and the visual neural signals are registered in the s1 slots of the respective sensory cortices and the stimulus pair is perceived as synchronous. **b** Dynamic-insertion mechanism of rapid temporal recalibration (TR). We denote $T_A$ the period of the fA cycle. In the example shown, the preceding trial (left) is of the t-1:V type, whereby the visual input precedes the auditory input. (Note that the model also applies to t-1:A trials.) In this t-1:V trial the SOA is such that the visual input is registered to a s1 slot and the auditory input to a s3 slot resulting in an asynchronous perception of the audiovisual stimulus pair. The Dynamic-insertion model predicts that during rapid temporal recalibration the phase occurrence of fA oscillatory segments along the slow cycles in the auditory and visual cortex is shifted to adapt to the current audiovisual modality order. The subsequent effect of this phase shift (highlighted with pink arrows) is that on the current trial (for illustration purposes t:V with similar SOA to trial t-1:V) both the auditory and visual stimuli are more likely to be registered to a s2 slot. The resulting subjective perception of the stimulus pair is more synchronous than on the previous trial. The audiovisual asynchrony on trial t-1 causes a shift in PSS, such that similar asynchronies on trial t appear less pronounced than before (as seen in behavioral data in Fig. 2b). **c** Empirical & PAC model of point of subjective simultaneity (PSS). The PSS values derived from the Dynamic-insertion model predictions (black bars) were statistically identical on average to those observed empirically in participants (gray bars), in both conditions (t-1:A, left; and t-1:V, right) with $p = 0.95$ and $p = 0.92$, respectively (two-sampled paired $t$ tests). Dots represent single-subject data ($n = 16$).

same comparison for the model data was not (PSS t-1:A vs. PSS t-1:V, $p = 0.17$, Bayes factor in favor of H0 = 1.65). These latter results indicate that the predictions from the model do not discriminate between conditions as clearly as behavior, yet with poor evidence from current sample that both model predictions are equivalent (small Bayes factor).

## Discussion
Our study reveals a neurophysiological mechanism of rapid, moment-to-moment adjustments of cortical dynamics for the perceptual realignment of temporally discrepant auditory and visual signals. This function is of crucial importance for adaptive behavior and social interactions. We demonstrate that in the auditory cortex and to some extent, the visual cortex, the coupling of high-frequency oscillations nested within an underlying alpha cycle accounts for temporal recalibration behavior. The parametric combination of frequency and phase of fast oscillations along the coupled slow-frequency cycle explains the sophisticated perceptual and behavioral adjustments observed empirically.

Our first main result shows that in the auditory cortex prior to stimulus onset, the period of fA oscillations is quantitatively

related to individual temporal recalibration behavior. The ratio between the amount of temporal recalibration and the period of fast oscillations was remarkably similar across individuals and approached quantitatively: $TR/T_A = TR \times fA \sim 1.5$, indicating the average amount of trial-to-trial shifts of the high-frequency segments along the underlying alpha cycle.

Our second main result addresses trial-to-trial phase shifts of fast-frequency oscillations along the slower alpha cycle directly. We show in the right auditory and visual cortex that phase-shift durations are contingent to context (i.e., the modality order of the stimulus pair presented in the preceding trial) and statistically identical to temporal recalibration behavior.

Our third main contribution is the *Dynamic-insertion* model of cortical registration of audiovisual inputs, which accounts for PSS and TR behavior. The model explains these behavioral observations with the parameters of neurophysiological PAC in sensory cortices over pre-stimulus baseline time periods.

Our results are aligned with the emerging view that the mechanistic role of fast oscillations coupled to low-frequency cycles is related to a form of temporal segmentation of sensory neural processing[25] (Fig. 1). The Dynamic-insertion model advances this notion to multimodal sensory integration. Our data show that following e.g., visual leads on the previous trial, stimulus processing in the auditory cortex is shifted on average by one and a half fA oscillatory cycle, thereby determining the amount of rapid temporal recalibration that takes place. This mechanism of discrete and flexible sampling of sensory inputs is most likely implemented by fast, inhibitory networks involving GABAergic interneurons[32], as evidenced by gamma-theta coupling in the hippocampus, where distinct spatial information is registered to distinct gamma sub-cycles nested in a theta cycle[18,20,33]. Jensen et al. [25] proposed a related framework based on PAC for temporal coding in the visual system based on alpha-to-gamma PAC observations. The high-frequency oscillations serve to segment and organize neuronal representations in time, thereby converting spatially distributed representations into a temporal phase code[25].

Our results suggest a prominent role of the auditory cortex for rapid temporal recalibration. The observed relationship between the fA period duration and temporal recalibration was restricted to the auditory cortex and was particularly prominent following visual leads on the previous trial. Stronger recalibration effects following visual leads were confirmed in our behavioral data (see Fig. 2b) suggesting an asymmetry in recalibration between visual and auditory leads. One possible interpretation is that leading cross-modal input from the visual cortex on the previous trial triggers rapid temporal recalibration in the auditory cortex. System asymmetries in the processing of audiovisual stimuli have been reported before. In speech processing for instance, although both cortices receive cross-modal input, visual speech processes modulate auditory cortex activity[34–36] and are predictable of speech sounds[37,38]. Accordingly, visual inputs are suggested to serve as alerting cues to the auditory cortex, priming corresponding neural circuits for optimal processing of incoming signals[37]. Our findings are therefore indicative that rapid temporal recalibration is a fast-acting, automatic mechanism that cues and prepares the auditory cortex for the upcoming speech signal.

Our data are also compatible with the notion that audiovisual asynchrony on the previous trial triggers the optimization of the timing of the high-frequency fA segment, for rapid temporal recalibration of the next expected discrepant visual and auditory signals. The dynamic-insertion model proposes that half of the period of a fast-oscillatory cycle instantiates a transient decrease of regional inhibition (Fig. 7a) at the expected time of arrival of sensory signals, creating an optimal time slot for the cortical registration of incoming sensory information. Subsequent fA half-cycles represent further cortical registration slots, although

with gradually elevated levels of regional inhibition. As such, dynamic insertion aligns with and extends the temporal phase code model.[25] each sensory registration slot represents an opportunity to optimally register a sensory event in time within sensory modalities; each slot also assigns an ordered value to sensory inputs. In the context of multimodal sensory integration, the Dynamic-insertion model predicts that subjective perception of the stimulus order in the stimulus pair is not based on the absolute time when the respective sensory signals reach their respective primary cortical regions. Instead, subjective perception depends on the rank of the slot where each sensory cortical input was registered to. Hence, if the auditory and visual inputs are registered to the first slot s1 of their respective target cortical regions, the subjective perception will be that of a pair of simultaneous sensory stimuli even though they were not physically generated synchronously (Fig. 7a, PSS example). Similarly, if the auditory and visual inputs are registered in different slots (e.g., s1 and s2), the stimulus pair will be perceived as asynchronous (Fig. 7a, SOA example). Following audiovisual asynchrony, temporal recalibration causes similar asynchronies on the next trial to be perceived as more synchronous than before, as observed behaviorally (Fig. 2b) and illustrated in Fig. 7b. Synchrony perception may become more ambiguous or incorrect if rapid temporal recalibration fails to optimize the opportunities for sensory registration along the alpha cycle, for sensory input will have to overcome gradually increasing levels of regional inhibition for successful registration.

How downstream decision brain circuits assign and process the ranked values of the respective audiovisual nested slots in sensory regions to yield subjective perception remains a fascinating question. Traditional models assume that the perceptual outcome reflects the relative time of arrival of sensory signals at a putative central brain site[39]. Accordingly, changes in perceived simultaneity would be accounted for by top-down modulations in processing speed in one modality[40,41]. We propose that when e.g., visual leads are expected, temporal recalibration shifts the visual cortex sensory registration slots in such a way that the next expected visual-led input be registered optimally—i.e. at a phase along the slow cycle where inhibition is reduced. Accurate temporal registration of the auditory input may then become less of a priority, for instance so that attentional resources be allocated to the first-detected sensory modality. Therefore, the auditory nested slot is shifted dynamically by one and a half high-frequency oscillatory cycle—i.e. at a phase angle along the slow cycle where inhibition is not reduced by a fast fA cycle. The outcome is that of a less precise perception of the actual time difference between sensory inputs, resulting in subjective reports of simultaneity, despite considerable physical asynchrony. As behavior shows, the number of synchronous reports indeed increases when the same sensory modality lead repeats from one trial to the next (Fig. 2b).

At this stage, we can speculate that the subjective perceptual outcome is issued by frontal and prefrontal brain circuits. The phase adaptation shown by our data and predicted by the Dynamic-insertion model in early unimodal brain areas is inspired by the concepts and previous empirical evidence of active inference in neural circuits[42,43]. The notion of active inference is similar to predictive coding[44], in the sense that internal representations of the sensory context and environment issue predictions concerning the expected nature and timing of upcoming physical sensory inputs. These predictions can be seen conceptually as top-down signals that modulate neurophysiological parameters of brain activity in early sensory areas—here, manifested as the fA phase shifts along the fP cycles of the Dynamic-insertion model. A possible rationale for this dynamic adaptation is outside the scope of the present report, but would be to enable

an organized form of temporal sampling of complex environmental conditions and optimize brain metabolic resources. Sensory regions compute a form of error signal accounting for the discrepancy between internal prediction signals and actual sensory inputs. These prediction error signals are likely to be conveyed back to frontal and prefrontal systems, for updating internal representation models and motivating behavior adaptation.

The observed PAC phase shifts along the alpha cycle were lateralized to the right hemisphere of the auditory and visual cortex. A possible reason may be functional asymmetries between the hemispheres, in particular for the auditory system[45–47]. Of note is the asymmetric sampling in time (AST) theory, which proposes that auditory cortices segment the initial input using different temporal windows. While the left auditory cortex integrates auditory signals preferentially into 20–50 ms segments corresponding to phoneme length, the right hemisphere integrates over 100–300 ms optimized for slower acoustic modulations[48]. This suggests that slow oscillations in the 3–10 Hz range dominate in the right hemisphere, while higher-frequency fluctuations in the range of 20–50 Hz are predominant in the left hemisphere. Previous findings demonstrated such spectral dissociation between left and right auditory cortex in their intrinsic oscillatory activity, as well as during sensory stimulation[45,46]. Here we investigated PAC along an underlying alpha oscillation of 9 Hz on average. According to AST, the lateralized modulation of coupling phases along the alpha cycle could be explained by the right hemisphere's preferred sampling rate. Hence, the right auditory cortex may be critical for optimizing sensory processing across longer temporal windows. The low-frequency component of PAC has been suggested to contribute to signal integration across larger distances, even across brain regions[16,17]. Via such mechanism, the right-lateralized phase effect in the auditory cortex may thus extend to visual regions for the functional integration of the audiovisual stimulus pairs presented.

Our present findings pertain solely to rapid temporal recalibration, which is adjusted on a trial-by-trial basis rather than after prolonged adaptation to asynchronous stimulus pairs. Although these two types of temporal recalibration may coexist[49], they appear to be independent and very likely rely on different underlying neural mechanisms. Rapid temporal recalibration allows for transient effects that follow the previous exposure to audiovisual asynchrony; the Dynamic-insertion model explains that shifts of fA along the alpha cycle can be implemented in a rapid and dynamic fashion. Prolonged adaptation, on the other hand, induces persisting effects that decline slowly through phase shifts of entrained neural oscillations[10]. The two types of temporal recalibration operate on different time scales and may combine to optimize multisensory integration.

Taken together, we believe the present contributions advance our mechanistic understanding of the neurophysiological foundations of multimodal sensory perception embedded in the fabric of time. This brain function has crucial implications to adjust behavior in ecological situations and social interactions. For this reason, the form of sensory integration proposed by the Dynamic-insertion model is likely to be impaired in neurological syndromes and mental health conditions, which we hope will inspire a thread of dedicated studies.

## Methods

**Participants**. Eighteen healthy human volunteers (8 females) between the ages of 20 and 45 participated in this study. All participants had self-reported normal hearing and normal or corrected-to-normal vision. None had any history of neurological or psychiatric disorders, and all completed the informed written consent form and MRI screening where applicable. The Montreal Neurological Institute Research Ethics Board approved all procedures.

**General procedure**. Participants first took part in a behavioral session lasting approximately 10 min to get accustomed to the task. They then underwent a 1.5 h MEG session comprising 20–30 min of subject preparation, 50 minutes of MEG recordings, and a 5-minute debriefing. For MEG source imaging purposes, we also obtained 1.5-T T1-weighted anatomical MRI scans from each participant.

**Stimulus presentation and behavioral task**. Participants engaged in an audio-visual simultaneity judgment task. Each trial started with the presentation of a white fixation cross at the center of a black screen. The audiovisual stimulus pair was presented after a variable pre-stimulus time interval, jittered between 2.3 and 2.8 s. The visual stimulus consisted of a white square presented at the center of the screen. It was presented for 3 video refresh durations amounting to 51 ms (refresh rate: 60 Hz; one refresh duration: 17 ms). The auditory stimulus was a 1000-Hz tone of the same duration presented binaurally through Etymotic ER-3A insert earphones with foam tips (Etymotic Research). Each participant adjusted the tone's volume to a comfortable sound level. The visual and auditory stimuli were presented at seven different SOA levels (0, ±100, ±150, ±175, ±200, ±225, ±350 ms; negative values indicate that the visual stimulus was lagging the auditory stimulus, positive values indicate the opposite). The participant's task was to determine whether the two stimuli were presented synchronously or not. The participant had two buttons available (one on each hand), where one was assigned "synchronous" responses and the other "asynchronous". The button assignments were randomized from one block to the next. The experimental recording session comprised 5 runs of 112 trials each (8 trials per SOA level/modality order), amounting to a total of 560 trials. Different trial types were presented at random.

All stimulus timings were verified during empty-room recordings using a photodiode attached to the presentation screen and a microphone attached to the ear tubes. During recordings, the audio signal was split and recorded as a channel in the data such that each stimulus onset could be precisely determined; a photodiode guaranteed precise recording of visual stimulus onsets.

**Behavioral data analysis**. Our experimental design assessed whether the mean scores from two experimental conditions (t-1:V versus t-1:A) were statistically different from one another. We generally used two-tailed, repeated-measures $t$ tests (also known as "paired samples" $t$ tests), or the Wilcoxon signed-rank test as a non-parametric alternative.

Similar to the results by [4], we found that recalibration was most pronounced for SOAs between 100 and 225 ms. Recalibration for SOAs of 0 ms ($t$ tests, $p = 0.7$) and ±350 ms ($p = 0.36/p = 0.1$) was not observed, likely because these SOAs were outside the range of perceived ambiguity. For all other asynchronies, at least one of the ± SOA pair revealed significant recalibration ($p \leq 0.01$). Consequently, to assess the neural mechanisms underlying rapid temporal recalibration, we excluded trials with 0 ms and 350 ms SOAs from subsequent analyses. All remaining trials were used for further analyses, irrespective of whether the participant's response was correct. In fact, we were particulary interested in the window of SOAs around the PSS where the participant's response was often wrong with the SOA perceived as synchronous despite physical asynchrony.

**MEG data acquisition**. Subject preparation consisted of taping 3 head-positioning coils on the participant's scalp (nasion, left preauricular, and right preauricular sites). The positions of the coils were measured relative to the participant's head using a 3-D digitizer system (Polhemus Isotrack). To facilitate anatomical registration with MRI, about 100 additional scalp points were also digitized. One pair of electrodes was positioned across the participants' chest to capture electrocardiographic (ECG) activity; another pair was attached above and below one eye to detect eye blinks and large saccades (EOG). All participants were measured in seated position using a whole-head 275-channel VSM/CTF system with a sampling rate of 2400 Hz (no high-pass filter, 660 Hz anti-aliasing online low-pass filter) in a magnetically shielded room (3-layer passive shielding). The head-positioning coils were energized prior to stimulus presentation to localize the participant's head with respect to the MEG sensors. Total head displacement was measured continuously and could not exceed 5 mm for inclusion in source analyses. Individual head-positioning locations were then used to coregister each participant's data to high-resolution T1-weighted MR anatomical images. A 2-minute empty-room recording performed immediately following data acquisition, with the same acquisition parameters, was used to capture sensor and environmental noise statistics. These latter were used for MEG source modeling, as explained below.

**MEG preprocessing**. All data preprocessing and MEG source imaging were performed using Brainstorm[50] [http://neuroimage.usc.edu/brainstorm]. All implementation details are readily documented and can be verified in Brainstorm's code. Data analysis was performed with Brainstorm and using custom Matlab scripts (The Mathworks Inc., MA, USA).

The MEG system's synthetic 3rd-order gradient balancing was used to remove background noise on-line. All recordings were visually inspected to detect segments contaminated by head movements or environmental noise sources, which were discarded from subsequent analysis. Heart and eye movement/blink contaminations were attenuated by designing signal-space projections (SSPs) from selected segments of data about each artifactual event. Using Brainstorm's ECG and EOG detection functionality, heartbeat events were automatically detected at

the R peak of the ECG's QRS complex, and eye blink events were determined automatically at the peaks of the EOG traces. Projectors were defined using principal component analysis (PCA) of these data segments filtered between 10 and 40 Hz (for heartbeats) or 1.5 and 15 Hz (for eye blinks) in a 160-ms time window centered about the heartbeat event, or 400 ms around the eye blink event. The principal component(s) that best captured the artifact's sensor topography were manually selected as the dimension(s) against which the data was orthogonally projected away. Note that in most participants, the first principal component was sufficient to attenuate artifact contamination. The projectors obtained for each participant were propagated to the corresponding MEG source-imaging operator as explained below. Power-line contamination (main and harmonics) was reduced by complex match filtering with 1-Hz resolution bandwidth for sinusoidal removal. Finally, the preprocessed data were resampled at 1200 Hz.

The scalp and cortical surfaces were extracted from the MRI volume data and a surface triangulation was obtained, all using FreeSurfer (surfer.nmr.mgh.harvard.edu) and subsequently imported into Brainstorm. The individual high-resolution cortical surfaces were down-sampled to about 15,000 triangle vertices to serve as image support for MEG source imaging.

**MEG source imaging**. Forward modeling of neural magnetic fields was performed using the overlapping-sphere technique implemented in Brainstorm[51]. In this method, one sphere is automatically adjusted locally to the individual scalp surface under each magnetic sensor to compute the corresponding lead field analytically. This method has been shown to provide the best trade-off between modeling precision and numerical accuracy[52]. The lead-fields were computed from elementary current dipoles distributed perpendicularly to the cortical surface from each participant. MEG source imaging was performed using the weighted minimum-norm estimator of the source time series.

**Event-related cortical responses**. We computed event-related responses across the cortex by averaging all trials time-locked to either the onset of the auditory or visual stimuli. Data were segmented in epochs of 0.7 s (200 ms pre- to 500 ms post-stimulus onset). Z-score normalization was applied using the baseline's (−200 ms to 0 ms) mean and variance. In order to determine the brain regions significantly activated above baseline, we computed parametric paired $t$ tests against the baseline. The resulting t-maps were evaluated at a threshold of $p = 0.01$. The performed tests were corrected for multiple comparisons via adjustment of the false discovery rate (FDR). Based on the single subjects' t-maps (highest significance at the evoked response's peak), we defined four regions of interest over left and right auditory and visual cortex, each comprising the same number of voxels ($n = 30$) resulting in comparable surface areas of 4-5 cm². From those regions of interest, we extracted the participants' average source time courses for conditions of interest to inspect activation across time. Average source time courses were z-score normalized using the epoch's baseline.

**Phase-amplitude coupling (PAC)**. 2 out of 18 participants were excluded from PAC analyses because a well-defined spectral peak necessary for the meaningful estimation of PAC was absent (see text for further details). PAC is defined as the modulation of a high-frequency oscillation's amplitude at frequency fA by the phase of a slower rhythm at frequency fP. We used the recently introduced tPAC (time-resolved phase-amplitude coupling) algorithm[30], which yields a time-resolved measure of PAC. Time variations of PAC were measured on 1-sec epochs centered at the onset of the first of the two stimuli (from −500 ms to +500 ms). A 500-ms sliding time window with an overlap of 50% was used to measure tPAC at 3 consecutive instances centered respectively at −250 ms, 0 ms, and +250 ms with respect to stimulus onset. Temporal recalibration across participants is typically estimated around 35 ms, which is compatible with oscillations in the range of 20–30 Hz. We thus defined the band of interest for fA to be within the human beta to gamma range (16-84 Hz); based on analysis of power spectral density (see Supplementary Fig. 1) the band and bins for fP were defined as 8–12 Hz (human alpha range). In subsequent analyses, the fA band was binned into 18 sub-bands of equal width. In brief, for estimation of tPAC in each fA sub-band, the recorded signal was filtered in that band, and its envelope ($Env_A$) extracted via the Hilbert transform. For detecting the slow oscillation the most coupled to these fast rhythms, the power spectrum of this envelope ($PS_{envA}$) and the power spectrum of the original recorded signal ($PS_{orig}$) in the alpha range were extracted. The most coupled frequency ($fP^*$) was defined as the one that had a maximum peak in the $PS_{envA}$ co-occurring with a peak in $PS_{orig}$. Then, the original input signal was filtered around this $fP^*$, and its phase ($\varphi_P$) was extracted using Hilbert transform. To estimate the coupling strength between the fA sub-band and $fP^*$, we defined the complex vector equal to $z(t) = Env_A \cdot e^{\varphi_P}$ for each time point. Euclidean average of these vectors in the current temporal window divided by the power of $Env_A$ provided a measure of coupling strength for the detected coupled frequencies. This procedure was repeated for all time windows, and all fA sub-bands. For each participant and ROI, the final PAC score was based on the combination of fP and fA sub-band that yielded the maximum PAC strength. The filters used for PAC analysis were even-order linear phase FIR filters, based on a Kaiser window design. The order was estimated using Matlab's *kaiserord* function and the filter generated with Matlab's *fir1*. Because the filters were linear phase, we compensated for the filter delay by shifting the filter application sequence backward in time by $M = N/2$ samples (Matlab's function *filtfilt*). This effectively makes the filters zero-phase and zero-delay. The resulting filtering process therefore did not smear the phase of signals in the time domain. We contained the edge ringing effects of filtering outside the PAC time window of interest, by designing a sufficiently long signal buffer on both sides of the analyzed epoch to contain 99% of the energy of the filter impulse response. Actual filtering was therefore performed on these longer epochs of the extended signal.

We further identified the phase of the low-frequency oscillation when the amplitude of the coupled high-frequency oscillations was highest. This resulted in measures of the PAC phase, for each participant, on each trial, and for each source vertex within a region of interest. For group analysis, we pooled all data and assessed whether phase distributions were identical between trial categories using the Kuiper test, a circular version of the non-parametric Kolmogorov–Smirnov test using the *Circular Statistics Toolbox for Matlab*[53]. For further analysis, we determined, for each participant and ROI, the cortical location (source vertex) that revealed the largest significant difference between trial categories. From each participant's single-trial phase data extracted at these locations we computed the participant's circular means and tested whether the measured phase angles were distributed non-uniformly for each trial category using Rayleigh tests. All tests were evaluated at alpha levels corrected for multiple comparisons ($p < 0.0125$).

Statistical significance of tPAC parameter estimates was verified with a non-parametric resampling technique. We followed the recommendations by[27], suggesting to generate surrogate datasets using a block-resampling approach, which was acknowledged to produce relatively conservative assessments of statistical significance of PAC measures. Briefly, the envelope time series of the fA oscillation in each time window were first split into five blocks. These blocks were then randomly permuted to yield a surrogate dataset that realizes the assumption of absence of PAC beyond chance levels. 500 surrogate trials were produced per time-window and fA center frequency candidate. The resulting tPAC and surrogate PAC distributions were compared using their Z-scores and evaluated at the critical values of $Z = 2.24$, which corresponds to Bonferroni-corrected significance levels (alpha) of 0.0125 in an upper-tailed Z-test.

For computation of Bayes factors, we used the following toolbox (https://klabhub.github.io/bayesFactor)[54].

**Reporting summary**. Further information on research design is available in the Nature Research Reporting Summary linked to this article.

## Data availability
The dataset generated and analyzed during the current study are available from the corresponding author upon reasonable request.

## Code availability
All code for MEG preprocessing, MEG Source Imaging and PAC analyses is readily available in *Brainstorm* (Version 2018).

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

## Acknowledgements
The authors thank Benjamin Morillon and Philippe Albouy for insightful comments on the manuscript. T.L. was supported by a CIHR Post-doctoral Fellowship. S.B. is grateful for the support received from the NIH (R01 EB026299), a Discovery grant from the Natural Science and Engineering Research Council of Canada (436355-13), the CIHR Canada research Chair in Neural Dynamics of Brain Systems, the Brain Canada Foundation with support from Health Canada, and the Innovative Ideas program from the Canada First Research Excellence Fund, awarded to McGill University for the Healthy Brains for Healthy Lives initiative.

## Author contributions
T.L. and S.B. designed the research, wrote the paper, and analyzed the data. T.L. performed the research. S.S. contributed to the PAC data analysis.

## Competing interests
The authors declare no competing interests.
