## [Peer Review File · Communications Biology]

Reviewers' comments:

Reviewer #1 (Remarks to the Author):

The present manuscript examines the physiological mechanisms related to audio-visual temporal recalibration – an adaptive process that serves to perceptually realign physically asynchronous events. Using MEG, the authors found that both the periods and phase-shifts of high-frequency oscillations nested within alpha oscillations may account for rapid inter-trial temporal recalibration behavior. Based on these results, the authors propose the Dynamic Integration model that explains the behavioral observations using parameters of phase-amplitude coupling and the duration of fast oscillations.

This is an important and timely topic, and the authors present interesting results that may lead to other studies testing their model's predictions. As the authors noted, the behavioral paradigm has been studied numerous times, but the physiological mechanisms were not yet investigated. The manuscript is well written, and I appreciate the rigorous analysis. I think that there are a few points detailed below that can improve the manuscript.

Specific comments:

1. Some parameters regarding the filters used for PAC calculation are missing (e.g. type and order of the filters). This is particularly important for the low frequency used to estimate the instantaneous phase, as that would influence the window that affects each time point in the analysis. Meaning, what is the influence from the post-stimulus window that 'smears' into the pre-stimulus window? Perhaps more importantly, is there any smearing from the response in trial t-1 to the pre-stimulus window in trial t? from figure 2, it seems that the responses do not drop back to baseline after 0.5 sec. Specifically, since the data was split based on trial t-1 (V or A), that means that the response profile of the event related responses will be different during the inter-trial-interval and might influence the low-frequency phase calculation. I do not feel strongly that this is the case here since the ITI is probably long enough to avoid contamination from previous evoked responses, but I think that should be directly addressed with emphasis on the exact window used, the differences in event related responses and the potential smearing.
2. Related to the previous comment - Page 13 – Regarding the phase shift - could this be affected by the previous evoked response? In t-1:V and t-1:A there will be a shift in the timing of the evoked response that might cause a phase shift when looking at each region.
3. One of the obvious concerns regarding the claims made in the manuscript is the fact that, while there is a trend towards higher pre-stimulus PAC, there is no significant increase in pre-stimulus PAC. The authors rely on visual inspection and the conservative statistics to claim that there are evidence supporting expression of PAC. Due to the absence of statistical validity, I suggest to at least discuss the implications of a potential lack of PAC increase on the analyses described later. The analysis also shows the percentage of trials that show a positive z-score for PAC. I wonder if there is some clustering of trial types (i.e. t-1:AV, t-1:VA, match, non-match) that tend to show higher z-scores for tPAC?
4. page 28 – The authors used the event-related cortical responses during the task as visual/auditory localizers to determine their regions of interest. Since the auditory/visual stimuli are very close to each other, aren't these time-locked ERF's contaminated by the other modality?
5. Did the authors only use trials in which the response was correct for tPAC and further analysis?
6. Page 11, para. 1-2 – the relationship between the high-frequency oscillations period and the individual temporal recalibration is an interesting finding. I wonder if this is specific to the period of the oscillations or could the authors also try a similar regression using the high-frequency power?
7. Page 12 (and supp. Figure S3) – I am not completely clear about Figure S3. The figure shows alpha power around t-1:A and t-1:V (indicated as time 0). However, the task-related analysis is concentrated around -500 ms – 0. If I understand correctly, we do not see this window (or at least not in its entirety) in Figure S3.
8. A previous MEG study (that is also mentioned in this manuscript; Kösem et al., 2014) tested temporal recalibration behavior after prolonged adaptation and found phase shifts of an entrained 1 Hz neural oscillations in auditory response that predicted the participants' shifts. Another behavioral study (Van der Burg et al., 2015) suggested that inter-trial recalibration and prolonged recalibration are independent of each other. Since the current study describes physiological mechanisms of inter-trial temporal recalibration, I suggest discussing possible mechanistic

differences between these two potentially independent types of temporal recalibration.

Additional comments:

9. Page 6, Results, para. 1 - "The SOA at the mode corresponds to the Point of Subjective Simultaneity (PSS), where two different sensory inputs are perceived as maximally simultaneous." - I think this sentence should be rephrased (?). Maybe "The SOA at the mode corresponds to the Point of Subjective Simultaneity (PSS), is defined as the point where two different sensory inputs are perceived as maximally simultaneous."
10. Figure 2 caption - (A) "The colored thin lines indicate the standard errors on the mean (SEM)." - since it refers to the dots, it might be better to put this sentence before the "Inset:". (B) The whiskers are referred to as "Black lines...", I think they are grey, not black.
11. Page 28 - "This method has been shown to provide the best trade-off between modeling precision and numerical accuracy." - I suggest adding a reference for this claim.
12. It might be interesting to discuss whether this process is automatic or does it require synchrony related task as introduced by the authors.
13. Based on their dynamic insertion model, could the authors speculate on what might determine the limits of perceived ambiguity?

Reviewer #2 (Remarks to the Author):

In the current manuscript, Lennert and collaborators use magnetoencephalography to study whether the temporal recalibration used to realign the inputs of two different modalities (i.e., auditory and visual) could be based on an adjustment of two oscillatory rhythms (i.e., slow and fast). This process of adjustment between neural oscillation rhythms is more commonly known as cross-frequency coupling. The authors present a series of strong, interesting and carefully presented arguments that pre-stimulus cortical rhythms in the sensory regions of the brain vary in a nested manner to allow A and V inputs to be perceived synchronously when the time lag is not too large. Overall, this is an excellent study: thorough, comprehensive, elegant, and theoretically well-motivated. The results are in many regards clear and support the conclusions of the study. Nevertheless, the methodological tour de force means that the main thread of the manuscript is somewhat lost. I have a few comments detailed below which I hope will help authors to further improve their manuscript.

Major.

1. My main point concerns the use of the comparison between the conditions t-1:V and t-1:A which is not sufficiently clear and consistent throughout the manuscript. Specifically, while Figure 1, which presents the experimental paradigm, does not mention this comparison, Figure 2 presents results that focus on this comparison. Then this comparison is dropped until Figure 5 where a crucial result on this comparison is highlighted. Finally, the comparison is not mentioned in the summary/explanatory figure of the results (Figure 7A/B).
 - a. Your experimental design is based on the comparison of two conditions t-1:V and t-1:A, an information that you highlight in Figure 2. Unfortunately, you do not state clearly why you expect a difference between the two modalities, and how it could be related to your hypothesis. Do you think that different cross-frequency coupling mechanisms could be at stake in these 2 conditions?
 - b. You do not specify whether the difference between PSS t-1:V and t-1:A (Figure 2A) is significant or not. I assume that it is not significant since you are presenting other comparisons in Figure 2B, but this needs to be clarified.
 - c. The latest comparisons presented in the manuscript (text related to Figure 7C) aim to show null differences between behaviour and neural activity in each condition (in PSS t-1:A and PSS t-1:V). Rather than looking for null results, could you test whether there is a significant difference between these two conditions, at the behavioural level? at the neural level (i.e., PAC)? An alternative analysis could be to compare the ratio sync/async between t-1:V and t-1:A, as you do for the behavioural data presented in Figure 2B?
 - d. The comparison PSS t-1:V and t-1:A is not mentioned in your explanatory/summary model (presented in Figures 7A and 7B), even though it underlies an important point in your argumentation.

2. I'm not sure to see the link between the fact that oscillations are involved in the segmentation of the incoming flow and their participation in the integration of 2 different modalities, as you stated on p.11: « Based on the previously proposed idea that PAC provides a temporal segmentation mechanism that discretizes continuous stimuli into smaller chunks, or slots, for further processing [23-25], we hypothesized that temporal recalibration is enabled by phase shifts of fA nested oscillatory segments (representing the "slots") along the slower fP oscillations". I think that a better justification is possible, or that you can unpack your idea to make it clearer.

Medium.

1. At the end of the introduction (P.5), you name the frequency bands that will be included in the experimental analysis. These choices (alpha, and beta/gamma) seem arbitrary in the introduction because it is not clearly stated that these choices are driven by the data - an information that I understood later when reading the manuscript. You should explicitly state in the introduction that you have no a priori assumptions about the frequency bands concerned, or you should describe previous results that support this selection.

2. p.6: "Temporal recalibration is defined as the difference between PSS of the t-1:V condition and the PSS of the t-1:A condition." Can you explain why temporal recalibration is reflected by the difference between the 2 conditions, and not only by the PSS of each condition? It seems to me that this is a crucial point in your argument, which could also respond in part to Major Point 1. In my opinion, the direction of the difference might carry crucial information but the relationship between the direction of the difference and the PAC mechanism is not clear.

3. The statistical power of this study is not reported: N = 18 in the behavioural measures and then N = 16 in the PAC measures is rather low. Your sample size seems unplanned, thus I suggest a post-hoc computation of the likelihood to observe a significant effect, given the current samples (see e.g.

<http://daniellakens.blogspot.com/2014/12/observed-power-and-what-to-do-if-your.html>)

4. p. 9: « We determined on a single trial basis the combination of fP and fA that yielded the maximum PAC strength [30]. » In the result section, it would be useful to indicate more clearly that the fP and fA only vary within a predefined frequency band. While reading the results section, I had the impression that you have left the definition of the frequency band completely free.

5. While Figure 3 presents results that clearly support the choice of the fP band (i.e., 8-12 Hz), the manuscript does not provide a clear justification for the choice of fA band frequencies. Figure 4B seems to justify the choice of the gamma band (based on one participant only, which seems rather insufficient), but not that of the beta band. The band fA is wide (i.e., 16-84 Hz). Is there any justification for this choice? The beta band is often functionally distinguished from the gamma band, which itself is differentiated into low gamma and high gamma. The concatenation of all these frequency bands deserves to be explained.

6. p. 11. "Depending on the nature of the stimulus pair on the preceding trial, nested high frequency cycles would be advanced or delayed along the underlying slow cycle." Could you unpack this idea? This is crucial information that could help the reader better understand the results.

Minor.

1. Figure 2B. "Black lines denote the interquartile range (quartiles Q1 and Q3)." The figure does not match the legend. Boxcars are indeed better to represent the dispersion than histograms. There is also an incompatibility between the text and the figure: "match" and "non-match" are in colors that do not correspond to the colors indicated in the text p. 7. It should also be made clear in the legend what is meant by match/non-match. This figure deserves to be better explained in the text as many comparisons are performed.

2. pp.7-8: "This result corroborates a central notion to temporal recalibration: previous exposure to a given amount of audiovisual asynchrony on trial t-1 causes a shift in PSS". If your result is in line with previous results, can you indicate the references? In my opinion, the shift in PSS is not such a common measure. You should be clearer about what has been shown in the literature and what your findings suggest.

3. On p.11, you use the terms "fA period": this term is a bit confusing as you introduced the relationship between "period TA" and fA in the previous paragraph.

4. Fig 5. In my version, auditory lead trials are presented in blue (and not in green as indicated in figure legend).

5. Fig 5. "The respective regression coefficients a , p -values, R^2 are indicated in the respective titles": You should use the word "panels" rather than "titles", and rephrase the sentence.
6. S3 Figure. Is it possible to plot the STD on top of the mean?

Signed: Sophie Bouton

We thank the Reviewers for their constructive feedback. We tried to address all concerns and questions as comprehensively as possible and updated the manuscript where indicated.

Reviewer #1 (Remarks to the Author):

The present manuscript examines the physiological mechanisms related to audio-visual temporal recalibration – an adaptive process that serves to perceptually realign physically asynchronous events. Using MEG, the authors found that both the periods and phase-shifts of high-frequency oscillations nested within alpha oscillations may account for rapid inter-trial temporal recalibration behavior. Based on these results, the authors propose the Dynamic Integration model that explains the behavioral observations using parameters of phase-amplitude coupling and the duration of fast oscillations.

This is an important and timely topic, and the authors present interesting results that may lead to other studies testing their model's predictions. As the authors noted, the behavioral paradigm has been studied numerous times, but the physiological mechanisms were not yet investigated. The manuscript is well written, and I appreciate the rigorous analysis. I think that there are a few points detailed below that can improve the manuscript.

Specific comments:

1. Some parameters regarding the filters used for PAC calculation are missing (e.g. type and order of the filters). This is particularly important for the low frequency used to estimate the instantaneous phase, as that would influence the window that affects each time point in the analysis. Meaning, what is the influence from the post-stimulus window that 'smears' into the pre-stimulus window? Perhaps more importantly, is there any smearing from the response in trial t-1 to the pre-stimulus window in trial t? from figure 2, it seems that the responses do not drop back to baseline after 0.5 sec. Specifically, since the data was split based on trial t-1 (V or A), that means that the response profile of the event related responses will be different during the inter-trial-interval and might influence the low-frequency phase calculation. I do not feel strongly that this is the case here since the ITI is probably long enough to avoid contamination from previous evoked responses, but I think that should be directly addressed with emphasis on the exact window used, the differences in event related responses and the potential smearing.

To address this Reviewer's concern about possible smearing effects from one analysis window to the next during PAC computation, we provided more information about the filters to the Methods section of the manuscript (page 32) and for details refer to our recent tPAC method paper (Samiee & Baillet, 2017). Briefly, the Brainstorm filters used for PAC analysis are even-order linear phase FIR filters, based on a Kaiser window design. The order was estimated using Matlab's kaiserord function and the filter generated with Matlab's fir1. Because the filters are linear phase, we compensated for the filter delay by shifting the filter application sequence backward in time by $M=N/2$ samples (Matlab's function filtfilt). This effectively makes the filters zero-phase and zero-delay. The resulting filtering process therefore did not smear the phase of signals in the time domain.

We contained the edge ringing effects of filtering outside the PAC time window of interest, by designing a sufficiently long signal buffer on both sides of the analyzed epoch to contain 99% of the energy of the

filter impulse response. Actual filtering was therefore performed on these longer epochs of extended signal.

This Reviewer also raised concerns about possible carry-over effects from one trial to the next. In order to counter such contamination, we purposely chose a very long ITI, rendering carry-over effects unlikely. Typically, the waveform and effects elicited by a stimulus can last for about one second (Woodman, 2010). In our case, there was more than 3 seconds between trials (2.3 – 2.8 sec ITI plus participant's response time). An ITI of such duration is longer than what typically implemented in similar tasks.

Fig 3B shows the time course of average event-related responses time-locked to the auditory and visual stimulus onsets. The figure displays only the first 500 ms of the post-stimulus response, hence the full return to baseline is not visible.

We extracted measures of pre-stimulus alpha power to verify there were no unwanted carry-over effects from the ERPs on the previous trial. The rationale for this approach was that alpha was the low frequency for phase used in present PAC derivations.

We did not find any significant difference in pre-stimulus alpha power in the LAC, RAC and RVC regions of interest, between experimental conditions. This argues against a form of low-frequency phase contamination in this frequency band where we reported key effects. The only significant difference in alpha power between trial types was found in LVC, where none of our findings (neither fA-related nor phase-related effects) pointed at.

We now provide a comprehensive illustration of these findings in Supplementary Figure S3.

We derived another control evaluation by assessing whether pre-stimulus PAC strength was similar across the study conditions. The estimation of PAC strength relies substantially on the signal component of PAC's low frequency for phase. In all tested ROIs, PAC strength was similar between t-1:A and t-1:V trials.

We provide these new results in the revised text (page 13) and as a new illustration in Supplementary Figure S4.

In sum, we trust these arguments and new derivations provide evidence there was no smearing effect of signals of interest in the time domain, neither across analysis time windows, nor across trials.

2. Related to the previous comment - Page 13 – Regarding the phase shift - could this be affected by the previous evoked response? In t-1:V and t-1:A there will be a shift in the timing of the evoked response that might cause a phase shift when looking at each region.

Building on our reply to the Reviewer's previous comment, we believe we now provide substantial evidence that the long ITIs of our study design prevents carry-over effects of the evoked response: at the beginning of the next trial, the event-related response to the previous trial's stimuli actually occurred more than 3 seconds before (2.3 – 2.8 sec plus participant's response time). An ITI of such a long duration is atypically long based on previously published similar tasks.

Further, event-related responses are relatively slow events. As such, the high-frequency PAC components (fA) we found in the beta-gamma range, shifting along the alpha cycle (fP PAC component), are not expected to be influenced by the event-related response on the previous trial.

While the phase shifts we reported are indeed influenced by the stimulus presentation of the previous trial (in line with our hypothesis), we believe our arguments above and our results demonstrate these effects are driven by an actual context-specific physiological mechanism, not a form of signal carry-over contamination from preceding event-related responses.

3. One of the obvious concerns regarding the claims made in the manuscript is the fact that, while there is a trend towards higher pre-stimulus PAC, there is no significant increase in pre-stimulus PAC. The authors rely on visual inspection and the conservative statistics to claim that there are evidence supporting expression of PAC. Due to the absence of statistical validity, I suggest to at least discuss the implications of a potential lack of PAC increase on the analyses described later. The analysis also shows the percentage of trials that show a positive z-score for PAC. I wonder if there is some clustering of trial types (i.e. t-1:AV, t-1:VA, match, non-match) that tend to show higher z-scores for tPAC?

We thank the Reviewer for this suggestion.

We now discuss in the main text (page 10) the possible reason for the absence of a clear statistical significance of pre-stimulus PAC in our data.

We argue that the absence of sensory inputs at this point in time in the trial may contribute to the weaker expression of PAC in auditory and visual cortices – two sensory brain regions whose activity strongly reflects the presence of stimulation. Participants at this point in time simply anticipated the presentation of a new stimulus pair, whose timing or category were impossible for them to predict.

When comparing PAC empirical statistics with surrogates, we did not analyse z-scores separately for the different trial types. We were indeed aiming to provide a general account of PAC in the regions of interest. Such analysis would be very substantially time consuming, and we are uncertain of its significance in the present context.

However, we wish to emphasize that a meaningful comparison indicator between trial types is pre-stimulus PAC strength. We compared the strength of PAC coupling prior to stimulus onset in both trial types but did not find significant differences.

We added this new data to the main text of the revised manuscript (page 13) and to a new Supplementary Figure S4.

This new result strongly suggests that PAC coupling (via PAC strength) was distributed evenly between experimental conditions.

4. page 28 – The authors used the event-related cortical responses during the task as visual/auditory localizers to determine their regions of interest. Since the auditory/visual stimuli are very close to each other, aren't these time-locked ERF's contaminated by the other modality?

We were not clear whether the Reviewer was referring to spatial or temporal proximity.

If spatial, we derived MEG source imaging based on the individual cortical surface from each participant (acquired using anatomical MRI). Hence, we believe the MEG imaging procedure was optimal, with

expected spatial resolution on the order of a few millimeters as shown in the context of high-resolution MEG retinotopic maps by Nasiotis et al. (2017).

We are therefore confident that there is very limited crosstalk between the event-related brain responses generated in the visual and auditory cortex, and that the statistical inference tests applied between conditions would further filter out such effect, as it is imposed by the laws of MEG physics, not by physiological fluctuations.

If temporal, short SOAs are not problematic for data analysis because the auditory evoked response is essentially contained in temporal cortex, while the visual evoked response localizes to occipital brain regions. Brain processing of the two sensory modalities is at this early, sensory stage separated into distinct auditory and visual streams, which can be readily disentangled anatomically.

In Figure 3, the absence of an event-related response in visual cortex following auditory stimulus onset (and vice-versa in auditory cortex) is evidence of the uni-modal nature of the responses measured in the auditory and visual regions of interest.

5. Did the authors only use trials in which the response was correct for tPAC and further analysis?

All trials were used for analyses, regardless of the participant's responses. In fact, we were particularly interested in the window of SOAs around the PSS (typically between -100 ms and +100 ms), where the participant's response is often wrong, as the SOA is perceived as synchronous despite physical asynchrony. These short SOAs were of particular interest as they reside in the range of audiovisual SOAs found naturally in speech, where temporal recalibration could be considered as naturalistically relevant. Van der Burg et al. (2014) showed in their original study that rapid temporal recalibration occurs regardless of whether the preceding trial was perceived as synchronous and whether a response was required or not.

It is thus believed that temporal recalibration is a fast-acting, sensory effect that occurs automatically, rather than driven by a high-level cognitive effect.

Thanks to this Reviewer, we have now clarified the nature of the trials that were used for the analyses performed in the revised Methods section (page 27,28).

6. Page 11, para. 1-2 – the relationship between the high-frequency oscillations period and the individual temporal recalibration is an interesting finding. I wonder if this is specific to the period of the oscillations or could the authors also try a similar regression using the high-frequency power?

We hypothesized that temporal recalibration would be related to the high-frequency period because of underlying shifts of those high-frequency oscillations along the low-frequency alpha cycle.

We did not have specific expectations regarding the high-frequency power with respect to temporal recalibration and hence decided to limit our analyses to the frequency's period.

How high-frequency power may be implicated in the process and mechanism of temporal recalibration would require is indeed a relevant and intriguing question that will require future work.

7. Page 12 (and supp. Figure S3) – I am not completely clear about Figure S3. The figure shows alpha power around t-1:A and t-1:V (indicated as time 0). However, the task-related analysis is concentrated around -500 ms – 0. If I understand correctly, we do not see this window (or at least not in its entirety) in Figure S3.

We apologize for this lack of clarity.

Figure S3 shows signal power in the alpha range before and during stimulus onset. Alpha power was critical in our analysis because alpha was designated as the low-frequency cycle of reference for PAC derivations.

The main point of the figure is to verify whether alpha power was modulated by the stimulus order on the previous trial. This is an important sanity check because systematic differences in alpha power could contaminate PAC strength or fA-fP coupling (with fP in the alpha range).

We found that alpha power was statistically identical for both trial types in all ROIs (evaluated at the multiple comparison corrected alpha level of $p < 0.0125$).

Alpha power was computed during a similar time window as our PAC derivations (-0.5 to 0.7 sec from stimulus onset). Because the resulting time-frequency decompositions are prone to edge effects, we cropped the time window at the edges for statistical analysis and plotting.

As pointed out by reviewer #2, we also added the standard errors to the plots.

8. A previous MEG study (that is also mentioned in this manuscript; Kösem et al., 2014) tested temporal recalibration behavior after prolonged adaptation and found phase shifts of an entrained 1 Hz neural oscillations in auditory response that predicted the participants' shifts. Another behavioral study (Van der Burg et al., 2015) suggested that inter-trial recalibration and prolonged recalibration are independent of each other. Since the current study describes physiological mechanisms of inter-trial temporal recalibration, I suggest discussing possible mechanistic differences between these two potentially independent types of temporal recalibration.

We thank the Reviewer for this suggestion. We have added a paragraph accordingly to the revised Discussion on page 25.

Additional comments:

9. Page 6, Results, para. 1 - "The SOA at the mode corresponds to the Point of Subjective Simultaneity (PSS), where two different sensory inputs are perceived as maximally simultaneous." – I think this sentence should be rephrased (?). Maybe "The SOA at the mode corresponds to the Point of Subjective Simultaneity (PSS), is defined as the point where two different sensory inputs are perceived as maximally simultaneous."

We thank the Reviewer for the suggestion and changed the sentence accordingly.

10. Figure 2 caption – (A) "The colored thin lines indicate the standard errors on the mean (SEM)." – since it refers to the dots, it might be better to put this sentence before the "Inset:".

(B) The whiskers are referred to as “Black lines...”, I think they are grey, not black.

Thank you: we have altered the text accordingly.

11. Page 28 – “This method has been shown to provide the best trade-off between modeling precision and numerical accuracy.” – I suggest adding a reference for this claim.

The reference provided one sentence later covers the above claim. We moved this citation in the text so it is now clearer to the reader (page 30).

12. It might be interesting to discuss whether this process is automatic or does it require synchrony related task as introduced by the authors.

We thank the Reviewer for the question and augmented the revised Discussion of the manuscript (page 22).

Briefly, we argue that rapid temporal recalibration is an automatic mechanism that is of ecological relevance in naturalistic situations processed by brain functions. Indeed, most naturalistic auditory and visual signals emanating from the same event – like speech for example – reach the brain with slightly different latencies as they are subject to small temporal processing delays.

We therefore argue that rapid temporal recalibration contributes to account for this timing difference based on context, as registered by the most recently experienced audiovisual asynchronies.

In natural settings, a given audiovisual asynchrony is likely to be stable for a while, based on current environmental conditions -- for instance in speech, when the distance and direction of speech sounds remain relatively constant during a conversation.

We do not anticipate that an experimental task as in the present study is necessary to induce temporal recalibration. Altering audiovisual asynchronies in a controlled manner, back and forth between successive trials allows to study rapid temporal recalibration in a systematic fashion.

Further, Van der Burg et al. (2014) showed in their original study that rapid temporal recalibration occurs regardless of whether the preceding trial was perceived as synchronous and whether a response was required or not. This underlines that temporal recalibration is a fast-acting, sensory effect that occurs automatically, rather than driven by high-level cognitive functions.

Finally, recent work (see references 7 and 8) demonstrated that rapid TR was altered in individuals on the autism spectrum, suggesting that the mechanism may be critical for normal multisensory temporal processing.

13. Based on their dynamic insertion model, could the authors speculate on what might determine the limits of perceived ambiguity?

The Dynamic Insertion Model (DIM) postulates that two stimuli are perceived as synchronous if the fA slot they are registered to is of the same rank. The farther apart the slots are in rank, the better they are perceived as asynchronous.

The DIM model further posits that there is an optimal phase along the slow alpha cycle for sensory inputs to be registered, that is, when alpha inhibition is the most reduced and fA modulation is the strongest. We hypothesized that a rapid temporal recalibration physiological mechanism shifts fA slots to optimize sensory processing.

If, however, optimization fails, registration of the sensory stimulus is less optimal because of higher alpha inhibition, resulting in a perception that is less precise.

This may lead to ambiguity in subjective perception or altered perception of the stimulus order.

We also argue that ambiguity about the perceived stimulus order may be issued further downstream in the course of the participant's decision process in response to the stimulus, when the timing of the sensory cortical inputs needs to be processed and transferred to a response decision concerning stimulus order, as in present task.

The Reviewer raised an important and meaningful point here: we expanded our conceptual arguments and emphasized the limits of synchrony perception within the notions of the proposed DIM model, in the revised Discussion (page 23).

Reviewer #2 (Remarks to the Author):

In the current manuscript, Lennert and collaborators use magnetoencephalography to study whether the temporal recalibration used to realign the inputs of two different modalities (i.e., auditory and visual) could be based on an adjustment of two oscillatory rhythms (i.e., slow and fast). This process of adjustment between neural oscillation rhythms is more commonly known as cross-frequency coupling. The authors present a series of strong, interesting and carefully presented arguments that pre-stimulus cortical rhythms in the sensory regions of the brain vary in a nested manner to allow A and V inputs to be perceived synchronously when the time lag is not too large. Overall, this is an excellent study: thorough, comprehensive, elegant, and theoretically well-motivated. The results are in many regards clear and support the conclusions of the study. Nevertheless, the methodological tour de force means that the main thread of the manuscript is somewhat lost. I have a few comments detailed below which I hope will help authors to further improve their manuscript.

Major.

1. My main point concerns the use of the comparison between the conditions t-1:V and t-1:A which is not sufficiently clear and consistent throughout the manuscript. Specifically, while Figure 1, which presents the experimental paradigm, does not mention this comparison, Figure 2 presents results that focus on this comparison. Then this comparison is dropped until Figure 5 where a crucial result on this comparison is highlighted. Finally, the comparison is not mentioned in the summary/explanatory figure of the results (Figure 7A/B).

The comparison between conditions t-1:V and t-1:A is at the core of the present study and integral to the definition of rapid temporal recalibration. From the Reviewer's comment above, we realize we might not have emphasized its significance clearly enough.

In the following, we explain the importance of the comparison between the two conditions and its role in the figures initially produced. We have also done our best to clarify these crucial elements in the main text of the revised manuscript and the updated figures.

Figure 1 displays the experimental paradigm highlighting the two types of stimulus modalities (auditory tone and visual flash) and the resulting three types of stimulus timings (visual lead $V < A$ (t:V), synchronous presentation (t:V=A), and auditory lead $A < V$ (t:A)).

We enhanced the revised Figure to illustrate both possible stimulus presentations on the preceding trial. The synchronous trial (t:V=A) is preceded by a visual lead trial (t-1:V), while trial #4 (t:A) is preceded by an auditory lead trial (trial #3). We hope this improves how the information is conveyed through the illustration of these two trial types. The actual comparison of behavioural data from t-1:A and t-1:V trials is in Figure 2.

Indeed in Figure 2A, we introduce the notion of rapid temporal recalibration (TR). TR is a measure of how much the point of subjective simultaneity (PSS) is shifted depending on whether the previous trial was a visual or auditory lead (that is, between t-1:V and t-1:A trials). In other words, temporal recalibration describes by how much simultaneity perception is recalibrated after a given asynchrony exposure. In order

to quantify the amount of this recalibration or shift, temporal recalibration is defined as the difference between the PSS of the t-1:V condition and the PSS of the t-1:A condition.

We clarified the importance of the definition of TR on page 6 of the revised manuscript.

In Figure 2B, we are again splitting t-1:V and t-1:A trials but we are also distinguishing between visual leads and auditory leads on the current trial. This means we are examining a more complex situation where all trial combinations are compared.

We hopefully improved the readability of the revised figure in that respect.

The main message from the figure is that for both visual and auditory leads on the current trial, the preceding trial influences whether the stimulus pair will be perceived as synchronous or not. In both cases the participant is more likely to perceive the pair as synchronous when it was preceded by the same stimulus order (that is, when there was a repeat in the order of stimulus presentation).

This corroborates that the PSS indeed shifts with respect to the stimulus order on the previous trial.

The second message is that this shift in PSS was particularly prominent when the preceding trial was a visual lead. In that case, the shift from perceiving the stimulus pair as asynchronous in auditory lead presentations (non-repeat) to perceiving it as synchronous for visual leads (repeats) was very pronounced. We believe this observation is significant and may be contributing to the stronger neural effects for t-1:V trials as shown in Figure 5.

We now discuss this result in greater depth in the revised version of the manuscript (pages 8 and 21,22).

In Figure 3 and Figure 4, we are showing the ROIs for the PAC analysis and illustrate the evidence of the expression of PAC in the empirical data. Here we are not distinguishing yet between the two trial types.

In Figure 5, we then study our first hypothesis, that is whether the participants' TR value (which is the shift in PSS between the two trial types) is systematically related to the participants' very own fA period.

Here we distinguish again between t-1:V and t-1:A trials to assess this relationship for both trial types.

In principle, it is possible that the relative shift in PSS between the two conditions could be a consequence of shifting in one condition while SOA remains stable in the other. Results shown in Figure 2B are compatible to this interpretation, where t-1:V trials may be more strongly modulated by trial context.

Indeed, while we found a positive relationship between TR and fA period is present in both conditions, the linear relationship is stronger in t-1:V trials. We revised the Discussion on page 21,22 to comment more thoroughly on this finding.

In Figure 6, we then examine the phase of PAC between the two conditions. The reasoning here is that TR – that is, the shift between the PSS of t-1:V and t-1:A – may be based on a relative shift of PAC along the cycle of the underlying alpha wave between the two conditions.

In other words, the phase of PAC in trials that were preceded by a visual lead presentation may be different to that of trials preceded by an auditory lead presentation.

Indeed, we found that this is the case (Figure 6B). Further, we show that the amount of phase shift is similar to the TR observed behaviourally (Figure 6C), which suggests that the two phenomena (in behaviour and neurophysiology) are directly linked.

Figure 7 proposes a model based on our results. We propose that TR is directly based on a relative phase shift of the coupling of fA oscillations along the slow alpha wave, which results in a relative shift of sensory processing of about the length of one fA cycle.

We found that this relative shift accounts for the behavioural changes experienced by our participants, in terms of synchrony perception (Fig 7C).

The Reviewer is correct that here we did not contrast directly between t-1:V and t-1:A trials. For the sake of simplicity and clarity of the example shown in Figure 7B, we present a t-1:V trial followed by a t:V trial to show a model of the predicted phase shifts at the neural level based on our results. However, the same mechanism shall apply to t-1:A trials followed by t:A trials.

We therefore clarified this important aspect in the revised Figure legend (page 18).

The example illustrates that following a t-1:V trial, the fA oscillations are shifted along the alpha cycle resulting in more synchronous perception of similar asynchronies on trial t. When stimulus orders do not repeat, as is the case for t-1:V trials followed by t:A or vice-versa, the mismatch likely resets rapid TR into the other direction.

Behaviourally, this would explain why we observed a clear distinction between t-1:V and t-1:A psychometric curves (Figure 2A) and changes in the ratio of asynchronous to synchronous responses (Figure 2B).

We hope that this outline clarifies the importance of the comparison of t-1:V and t-1:A trials for TR and emphasize the consistency of our analyses across conditions.

a. Your experimental design is based on the comparison of two conditions t-1:V and t-1:A, an information that you highlight in Figure 2. Unfortunately, you do not state clearly why you expect a difference between the two modalities, and how it could be related to your hypothesis. Do you think that different cross-frequency coupling mechanisms could be at stake in these 2 conditions?

Our first hypothesis is based on the idea that PAC provides a temporal segmentation mechanism that discretizes continuous stimuli into slots for further processing.

We hypothesized that rapid temporal recalibration is enabled by phase shifts of fA nested oscillatory segments (representing the “slots”) along the slower fP oscillations. Depending on the asynchrony of the stimulus pair on the preceding trial, nested high frequency cycles would be advanced or delayed along the underlying slow cycle. Consequently, signal processing is shifted to a previous or subsequent slot – e.g., by one high-frequency fA oscillatory cycle or more – thereby determining in a flexible manner the magnitude of temporal recalibration that takes place.

Thus, we first assess whether there is a linear relationship between the TR estimates from behavior and the subject-specific fA period (i.e., cycle length).

We expected this relationship to be similar for t-1:V and t-1:A trial types because we anticipated the subject-specific high-frequency oscillation (fA) to be the same across conditions.

We nonetheless distinguished between trials, as our behavioural data suggests that t-1:V trials may be more strongly modulated by the modality order on the preceding trial. Indeed, while the fA oscillation is not significantly different between trial types, we did observe a stronger linear relationship for t-1:V trials in auditory cortex compared to t-1:A trials.

Concerning our second hypothesis though, we did expect to report a difference in PAC phase between the t-1:V and t-1:A conditions. We predicted that it is the phase of the coupling to the alpha cycle that differs between conditions and promotes the neural basis of TR.

In other words, we postulated that a shift in fA-to-fP coupling phase between conditions enables rapid temporal recalibration.

In order to clarify these hypotheses and the role of the comparison of t-1:V and t-1:A trials, we now provide more detail and explanations to the revised version of the main text (pages 11 and 13).

b. You do not specify whether the difference between PSS t-1:V and t-1:A (Figure 2A) is significant or not. I assume that it is not significant since you are presenting other comparisons in Figure 2B, but this needs to be clarified.

The difference between PSS values is significant. A paired samples t-test yields a t-value of 5.275; corresponding to a p-value of 0.00009.

We added this information to the revised main text (page 6).

The comparisons in Figure 2B contribute to illustrating this effect and other aspects of the behavioural data. They account for the ratio of asynchronous-to-synchronous reports while taking into account the context of the current trial (that is, the modality order on the previous trial as well as on the current trial).

c. The latest comparisons presented in the manuscript (text related to Figure 7C) aim to show null differences between behaviour and neural activity in each condition (in PSS t-1:A and PSS t-1:V). Rather than looking for null results, could you test whether there is a significant difference between these two conditions, at the behavioural level? at the neural level (i.e., PAC)? An alternative analysis could be to compare the ratio sync/async between t-1:V and t-1:A, as you do for the behavioural data presented in Figure 2B?

We followed the Reviewer's suggestion and compared the two conditions with the behavioural data level and the predictions from the DMI model.

The behavioral data showed a significant difference between conditions (PSS t-1:A vs. PSS t-1:V, $p < 1e-4$), not the DMI model (PSS t-1:A vs. PSS t-1:V, $p = 0.17$) although with a small Bayes factor of 1.65, indicating poor evidence in favor of H0 (no difference between conditions).

These results indicate that the DMI model does not discriminate between conditions as clearly as behavior. We included these new results in the revised main text of the manuscript (page 20).

We are grateful to the Reviewer for their contribution to a more complete account of the data.

In that spirit, we now also report Bayes factors for testing the statistical equivalence between the predictions from the DMI model and behaviour per study condition. In the PSS t-1:A condition, we obtained a Bayes factor (BF) of 3.91, which indicates substantial evidence for the equivalence between the DMI model and behavior ($p = 0.95$). In the PSS t-1:V condition, the Bayes factor was 3.90, which also points at substantial statistical equivalence between behaviour and the DMI model predictions ($p = 0.92$).

We added this new result to the revised text of the manuscript (page 20).

We also looked into the comparison between the ratio of async/sync responses, following the recommendation of the Reviewer, but unfortunately, this latter is not possible because the neural data was not structured and tagged accordingly.

d. The comparison PSS t-1:V and t-1:A is not mentioned in your explanatory/summary model (presented in Figures 7A and 7B), even though it underlies an important point in your argumentation.

The Reviewer is correct that we did not contrast directly t-1:V against t-1:A trials in this figure. Our motivation was to favor simplicity and clarity of the presentation. We show in Figure 7B a t-1:V trial followed by a t:V trial because, the same sensory modality order in two successive trials elicited the strongest temporal recalibration behavioral effect. We used this example to illustrate the DIM phase shifts at the neurophysiological level, inspired by behavioral observations. We emphasize however that the same mechanism is expected to apply to situations when t-1:A trials are followed immediately by t:A trials.

We thank the Reviewer for giving us the opportunity to clarify this important point in the revised legend of Figure 7 (page 18).

2. I'm not sure to see the link between the fact that oscillations are involved in the segmentation of the incoming flow and their participation in the integration of 2 different modalities, as you stated on p.11: « Based on the previously proposed idea that PAC provides a temporal segmentation mechanism that discretizes continuous stimuli into smaller chunks, or slots, for further processing [23-25], we hypothesized that temporal recalibration is enabled by phase shifts of fA nested oscillatory segments (representing the “slots”) along the slower fP oscillations”. I think that a better justification is possible, or that you can unpack your idea to make it clearer.

The proposed Dynamic Integration Model predicts that rapid temporal recalibration operates by shifts of the fA oscillatory bursts along the underlying alpha cycle, to optimize sensory processing in accordance with the most recent asynchrony exposure. If the stimuli are registered to fA slots of the same rank within their respective cortical regions, they are perceived as synchronous. In other words, temporal recalibration modulates the timing of the processing of the auditory and visual stimuli and slightly shifts the processing of one modality with respect to the other to account for the most recently perceived audiovisual asynchrony.

The focus of the present was to clarify the mechanistic nature of neural mechanisms in early cortical areas that enable rapid temporal recalibration. We believe our empirical data provide substantial evidence that DIM represents a significant neurophysiological mechanism of multimodal sensory registration, that facilitates the actual integration of two sensory modalities by downstream brain circuits. Further studies, including our own future work, are required to replicate these findings and clarify the nature of the operations performed by higher-order brain regions to integrate DIM signals.

At this stage, we can speculate that the subjective perceptual outcome is issued by frontal and prefrontal brain circuits. The phase adaptation shown by our data and predicted by the DIM model in early unimodal brain areas is inspired by the concepts and previous empirical evidence of active inference in neural circuits (e.g., Schroeder et al., 2010; Donhauser and Baillet, 2020). The notion of active inference is similar to predictive coding, in the sense that internal representations of the sensory context and environment

issue predictions concerning the expected nature and timing of upcoming physical sensory inputs. These predictions can be seen conceptually as top-down signals that modulate neurophysiological parameters of brain activity in early sensory areas – here, manifested as the fA phase shifts along the fP cycles of the DIM model. A possible rationale for this dynamic adaptation is outside the scope of the present paper but could enable an organized form of temporal sampling of complex environmental conditions and optimize brain metabolic resources. Sensory regions compute a form of error signal accounting for the discrepancy between internal prediction signals and actual sensory inputs. These prediction error signals are likely to be conveyed back to frontal and prefrontal systems, for updating the self’s internal representation models and behaviour adaptation.

We now underline these conceptual aspects further in the revised Discussion, for inspiration of future research.

Medium.

1. At the end of the introduction (P.5), you name the frequency bands that will be included in the experimental analysis. These choices (alpha, and beta/gamma) seem arbitrary in the introduction because it is not clearly stated that these choices are driven by the data - an information that I understood later when reading the manuscript. You should explicitly state in the introduction that you have no a priori assumptions about the frequency bands concerned, or you should describe previous results that support this selection.

We thank the Reviewer for pointing this out.

We rephrased our statements in the revised Introduction accordingly.

We now indicate explicitly pp 4-5 that our core hypothesis was that PAC may enable a form of optimized registration mechanism of (sensory) inputs to brain circuits. We believe the references cited before that paragraph provide conceptual and some early empirical foundations to this assumption. Further, we moved reference [25] to the end of this revised paragraph, to acknowledge and emphasize more explicitly previous and similar viewpoints in the field. We also expanded the last paragraph of p. 5 accordingly.

2. p.6: “Temporal recalibration is defined as the difference between PSS of the t-1:V condition and the PSS of the t-1:A condition.” Can you explain why temporal recalibration is reflected by the difference between the 2 conditions, and not only by the PSS of each condition? It seems to me that this is a crucial point in your argument, which could also respond in part to Major Point 1. In my opinion, the direction of the difference might carry crucial information but the relationship between the direction of the difference and the PAC mechanism is not clear.

Rapid temporal recalibration is a measure of how much the point of subjective simultaneity (PSS) is shifted between t-1:V and t-1:A trials. In other words, TR describes by how much simultaneity perception is recalibrated after a given asynchrony exposure. In order to quantify the amount of this recalibration or shift, TR has to be defined as the difference between the two PSS values.

The direction of the difference determines the sign: the difference between PSS t-1:V and PSS t-1:A results in a positive TR; the other way round yields a negative TR. A negative TR is not meaningful in this context; it is the absolute difference in time between the two PSS values that is the measure of interest.

Although TR occurs in both directions, from PSS t-1:V toward PSS t-1:A, and vice-versa, it is possible that the strength of temporal recalibration is not symmetric in both directions. Indeed, our data suggest that temporal recalibration may be stronger following t-1:V trials. However, this does not change the actual definition of TR.

We clarified this issue with a revision of the corresponding Results section (page 6).

3. The statistical power of this study is not reported: N = 18 in the behavioural measures and then N = 16 in the PAC measures is rather low. Your sample size seems unplanned, thus I suggest a post-hoc computation of the likelihood to observe a significant effect, given the current samples (see e.g. <http://daniellakens.blogspot.com/2014/12/observed-power-and-what-to-do-if-your.html>)

We agree with the Reviewer that statistical power would increase with more participants. promoting the sensitivity of statistical inference by decreasing the probability of a Type-II error.

For running linear regression, a common practical recommendation for detecting reasonably sized effects with reasonable power is to use at least 10-20 observations per parameter (Harrell, 2001). With n=16, we believe we are well within that recommended range. Using an online sample size calculator (<http://www.sample-size.net/correlation-sample-size/>) with an alpha error rate of 0.004 (corrected for multiple comparisons), a beta of 0.3, and an expected effect size of 0.7, a sample size of n=18 is suggested. Lowering the effect size from strong to medium increases the sample size to > 25 indicating that we may not have the statistical power to detect effects of medium and low strengths with our sample size of 16. Because the correlation effects in auditory cortex were very strong, with effect sizes of >0.7, we believe that our conclusions are justified.

However, as typical of experimental reports, we agree with the Reviewer that other possible effects of lesser strength may not have been detected with our present data sample. As mentioned above, we believe the novelty and significance of the present findings and their proposed mechanistic interpretation will inspire future studies, with other specific aims and appropriate sample sizes.

4. p. 9: « We determined on a single trial basis the combination of fP and fA that yielded the maximum PAC strength [30]. » In the result section, it would be useful to indicate more clearly that the fP and fA only vary within a predefined frequency band. While reading the results section, I had the impression that you have left the definition of the frequency band completely free.

We clarified these aspects in the revised Results and Methods sections, with rewritten definitions of the fA and fP frequency bands (pages 8,9,31).

5. While Figure 3 presents results that clearly support the choice of the fP band (i.e., 8-12 Hz), the manuscript does not provide a clear justification for the choice of fA band frequencies. Figure 4B seems to justify the choice of the gamma band (based on one participant only, which seems rather insufficient), but not that of the beta band. The band fA is wide (i.e., 16-84 Hz). Is there any justification for this choice? The beta band is often functionally distinguished from the gamma band, which itself is differentiated into low gamma and high gamma. The concatenation of all these frequency bands deserves to be explained.

Our reasoning for choosing the beta and gamma bands as fA frequencies of interest was based on our hypothesis that subject-specific, fast oscillations nested in slower cycles are related to TR and can account for the observed TR estimate. In the literature, the reported average amount of rapid temporal recalibration is around 30 ms. Our own behavioural results confirmed temporal recalibration in that range across subjects (see Figure 2A).

Frequencies in the beta and gamma range have periods compatible with these estimates. According to our hypothesis, a participant with a TR of 30 ms would reveal PAC with fA of around 33 Hz; someone with a TR of 50 ms would show PAC with fA of around 20 Hz, etc.

We thus used a frequency range of interest for possible fAs that contained beta and gamma frequencies (16-84 Hz). For PAC computation, the tested ranges for fP and fA were divided into bins (8/9/10/11/12 Hz for fP; 16/20/24/28/.../80/84 Hz for fA). PAC scores were then computed for each combination of fP-fA sub-bands. The fP-fA combination associated with the largest PAC score determine this participant's PAC statistic.

We clarified our reasoning for choosing these frequency ranges of interest in the revised main text (pages 8,9) and Methods section (page 31).

6. p. 11. "Depending on the nature of the stimulus pair on the preceding trial, nested high frequency cycles would be advanced or delayed along the underlying slow cycle." Could you unpack this idea? This is crucial information that could help the reader better understand the results.

The main idea behind rapid temporal recalibration is that the processing of auditory and visual sensory signals is shifted (advanced or delayed) depending on the most recent asynchrony exposure. Specifically, the relative timing of audio and visual signal processing is shifted in such a manner that similar audiovisual asynchronies on the next trial appear more synchronous than before.

The Dynamic Insertion Model provides a framework for this mechanism by proposing that the high frequency (fA) that in PAC is nested to a slow alpha frequency (fP) provides time slots of lower alpha inhibition where incoming stimuli can be more readily registered. By shifting the nested fA oscillations along the alpha cycle, processing of the neurophysiological signals from one of the sensory modalities can effectively be advanced or delayed.

The core novelty of the manuscript is that we propose that this defines the essential neurophysiological nature of rapid temporal recalibration and leads to modulations of synchrony perception in behavior.

We clarified this key element of our report further, by expanding the main text of the manuscript accordingly (page 11).

Minor.

1. Figure 2B. "Black lines denote the interquartile range (quartiles Q1 and Q3)." The figure does not match the legend. Boxcars are indeed better to represent the dispersion than histograms. There is also an incompatibility between the text and the figure: "match" and "non-match" are in colors that do not correspond to the colors indicated in the text p. 7. It should also be made clear in the legend what is meant by match/non-match. This figure deserves to be better explained in the text as many comparisons are performed.

We acknowledge that using the *match/non-match* wording for descriptors have caused confusion. We now refer to these latter as “repeat” and “non-repeat” and clarified their respective definitions in more detail in the revised main text and figure legends.

The mismatches between colors used in the Figure 2B and the colors indicated in the text have been corrected: thank you!

2. pp.7-8: “This result corroborates a central notion to temporal recalibration: previous exposure to a given amount of audiovisual asynchrony on trial t-1 causes a shift in PSS”. If your result is in line with previous results, can you indicate the references? In my opinion, the shift in PSS is not such a common measure. You should be clearer about what has been shown in the literature and what your findings suggest.

Indeed, the shift in PSS we observed is in line with previous results. A number of studies demonstrated that repeated exposure to an audiovisual asynchrony shifts the PSS in the direction of the leading sense (Fujisaki et al. 2004, Vroomen et al. 2004) as well as following single exposure to audiovisual asynchrony (van der Burg et al. 2013, van der Burg and Goodbourn, 2015, Simon et al., 2017, Simon et al., 2018). A shift in PSS is in fact considered one of the hallmarks of temporal recalibration and how this latter is defined.

We expanded the corresponding citations in the revised main text to underline the significance of our claims and findings (page 8).

3. On p.11, you use the terms “fA period”: this term is a bit confusing as you introduced the relationship between “period TA” and fA in the previous paragraph.

Thank you for pointing that out. We changed the wording to make it less confusing.

4. Fig 5. In my version, auditory lead trials are presented in blue (and not in green as indicated in figure legend).

We are sorry about this confusion: we have now updated the colors in the revised manuscript.

5. Fig 5. “The respective regression coefficients a, p-values, R2 are indicated in the respective titles”: You should use the word “panels” rather than “titles”, and rephrase the sentence.

Thank you: we have changed the wording in the revised manuscript.

6. S3 Figure. Is it possible to plot the STD on top of the mean?

We thank the reviewer for pointing this out. We added standard errors of the mean to the plots.

Signed: Sophie Bouton

References:

Nasiotis, K., Clavagnier, S., Baillet, S. and Pack, C.C., 2017. High-resolution retinotopic maps estimated with magnetoencephalography. *Neuroimage*, 145, pp.107-117.

Harrell, F.J. (2001) *Regression Modeling Strategies*. Springer

REVIEWERS' COMMENTS:

Reviewer #1 (Remarks to the Author):

The authors provided a considerable revision and addressed the major concerns previously raised. The manuscript is significantly improved and I do not have any further comments.

Signed: Idan Tal, PhD

Reviewer #2 (Remarks to the Author):

The authors have responded to all my comments. I think the paper is now well suited to the journal Communications Biology and can be accepted for publication.